# Enabling Detailed Action Recognition Evaluation Through Video Dataset Augmentation

**Jihoon Chung**
Princeton University
jc5933@princeton.edu

**Yu Wu**
Princeton University
yuwu@princeton.edu

**Olga Russakovsky**
Princeton University
olgarus@princeton.edu

## Abstract

It is well-known in the video understanding community that human action recognition models suffer from background bias, i.e., over-relying on scene cues in making their predictions. However, it is difficult to *quantify* this effect using existing evaluation frameworks. We introduce the Human-centric Analysis Toolkit (HAT), which enables evaluation of learned background bias without the need for new manual video annotation. It does so by automatically generating synthetically manipulated videos and leveraging the recent advances in image segmentation and video inpainting. Using HAT we perform an extensive analysis of 74 action recognition models trained on the Kinetics dataset. We confirm that all these models focus more on the scene background than on the human motion; further, we demonstrate that certain model design decisions (such as training with fewer frames per video or using dense as opposed to uniform temporal sampling) appear to worsen the background bias. We open-source HAT to enable the community to design more robust and generalizable human action recognition models. [1]

## 1 Introduction

Human action recognition is about understanding what the *human* in the video is doing; however, human action recognition models frequently rely on background cues to make their predictions. Prior works [6, 33, 59, 60, 71] have leveraged visualization tools like GradCam [46] to demonstrate that the video background significantly influences the prediction of human action recognition models. This occurs due to representation bias in the dataset, where particular actions (e.g., eating) tend to occur in particular environments (e.g., kitchens). Such concerns limit the practical usability and generalizability of models despite the impressive overall progress in the field [36, 61, 67].

While it is known that this background bias phenomenon is occurring, *quantifying* the degree to which it is occurring is still necessary. Being able to accurately assess how much human action recognition models rely on human features rather than background scene cues would allow researchers to compare different model designs and select the ones that would be robust to their unique test domains. Efforts such as [34, 63] have introduced datasets for quantifying background bias; however, scaling up their approaches may be prohibitively expensive due to the reliance on manual annotation.

In this work, we introduce the Human-centric Analysis Toolkit (HAT) to measure background bias in human action recognition models without the need for costly human annotation. We leverage recent

---

[1] https://github.com/princetonvisualai/HAT

36th Conference on Neural Information Processing Systems (NeurIPS 2022) Track on Datasets and Benchmarks.

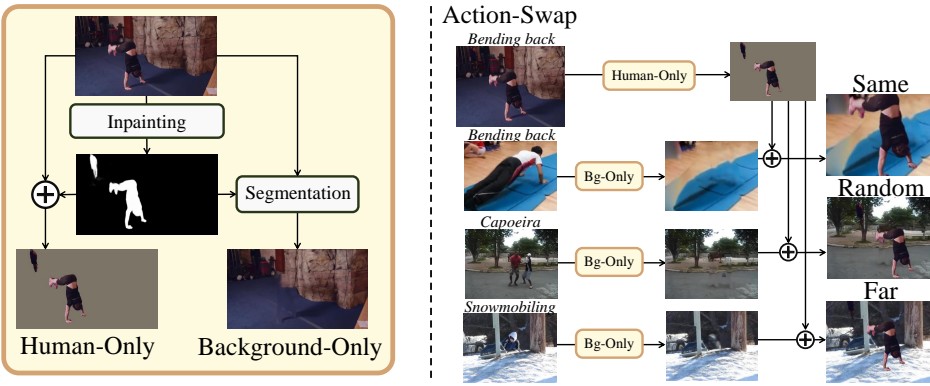

Figure 1: The pipeline of our Human-centric Analysis Toolkit (HAT). **Left:** HAT takes a video, segments the spatio-temporal human figure, and generates the Human-Only and Background-Only videos. **Right:** HAT generates Action-Swap videos by pasting the same human figure onto the Background-Only video from the same, a randomly-selected, and a far (dissimilar) action class.

improvements in image segmentation [24, 32, 38] and video inpainting [29, 35, 37] to automatically synthesize counterfactual videos containing Human-Only (a spatio-temporal segmentation of the human figure against a gray background), Background-Only (the video with the human removed via inpainting) or Action-Swap (human figure against an unusual background). Examples are shown in Figure 1. This process is efficient and scalable, requiring no manual annotation. HAT thus enables us to evaluate the sensitivity of human action understanding models to the different visual cues by comparing the accuracy on the original and synethetically manipulated videos.

We demonstrate the capabilities of HAT by running extensive analysis of human action recognition models trained on the Kinetics-400 [28] dataset. Concretely, we evaluate 74 trained models, corresponding to 14 different model designs (TSN [61], I3D [5], Non-local Neural Networks [62], R(2+1)D [54], TSM [36], SlowFast/SlowOnly [15], CSN [58], TIN [49], TPN [68], X3D [14], OmniSource [12], TANet [39], and TimeSformer [4]) with varying hyperparameters and backbone architectures provided by the MMAction2 [8] implementation. Some of our findings include:

- All 74 models exhibit strong background bias. When evaluated on the Action-Swap videos, the 74 models predicted the action class of the human 16.8% of the time on average – but predicted the action class of the randomly-selected background 29.5% of the time!

- Models trained with fewer frames per video appear to be more prone to background bias. For example, the TSN-based models [61] trained with 8, 5 and 3 frames per video retain 0.679, 0.683 and 0.694 of their original accuracy respectively when evaluated on the Background-Only videos, demonstrating consistently high and somewhat *increasing* background bias.

- Models trained with dense temporal sampling around a single timestep appear to be more prone to background bias compared to models trained with uniform sampling throughout the video. For example, when evaluated on the Background-Only videos as above, TSM-based models [36] with dense sampling exhibit strong background bias by retaining 0.703 of the original accuracy compared to only 0.675 with uniform sampling.

Overall, we make three contributions. First, we develop and open-source the Human-centric Analysis Toolkit (HAT), which generates synthetic videos to evaluate the background bias learned by human action recognition models. Second, we demonstrate its capabilities through extensive evaluation of 74 released models. Finally, we show that HAT can identify the design choices that appear to influence the amount of background bias learned by the model, helping inform future model design.

## 2    Related Work

**Human Action Recognition Models.** Currently, human action recognition is largely dominated by deep learning methods. With strong success in image-based tasks [10, 20, 30, 51], CNN-based deep learning models [12, 14, 15, 36, 39, 49, 54, 58, 61, 62, 68] were the go-to method for human action recognition, with gradual improvements in the model structure going from 2D-CNN [20, 44, 61]

to 3D-CNN [5, 14, 15, 54] to CNN models with specific temporal modeling [36, 39, 49]. A recent trend [2, 4] in human action recognition is to use a transformer module [57] as it has shown good performance [11] in image-based tasks. Another trend [4, 12, 58] is to incorporate large-scale datasets [16, 26, 65] into the training. In this work, we evaluate multiple action recognition models [4, 5, 12, 14, 15, 36, 39, 49, 54, 58, 61, 62, 68] in an effort to identify design decisions which appear to correlated with learned background bias.

**Human Action Recognition Dataset.** Early datasets [45, 66] offered a handful of human action classes that were collected in a controlled environment. UCF101 [52] and HMDB51 [31] were some of the first few datasets that were suitable for machine learning tasks. Although there are many different human action datasets [7, 9, 19, 47, 48, 73], the most popular dataset must be Kinetics-400 [28], due to its large size and variety of actions. However, due to the cost of collecting video datasets, the size of the dataset is still smaller than image datasets. Synthetic datasets [13, 17, 27, 41, 53], often used mixed with the real dataset, are popular methods of collecting data in an affordable manner. In human action recognition, the synthetic datasets are often used for training dataset [55, 56], and the model is tested on real videos. In this work, we generate synthetic counterfactual videos to enable detailed model evaluation without the need for costly annotation.

**Human-centric Analysis.** As the models have grown more complex, there has been an increased need for frameworks that provide insights into the model behavior beyond just a single accuracy number. Efforts have included model interpretability techniques [3, 46], detailed error analysis using additional manual annotations [1, 22, 43, 50], and (recently) stress-testing using automatically generated text or image data [25, 42]. There are a number of works studying specifically the impact of the human figure on human action recognition models. A common strategy employed by [6, 33, 59, 60, 71] is to use the GradCam [46] visualization to qualitatively demonstrate that the model's attention is on the background cues rather than on the human in the video. Several of these works [6, 59, 60, 71] propose methods to mitigate the effects of background bias during training; they evaluate its success both qualitatively through GradCam and quantitatively via accuracy on a downstream action recognition task (after fine-tuning the model trained with their new background-debiasing method). While this successfully demonstrates that their innovation is effective for model pre-training, it does not directly measure the learned background bias. The most natural analysis is to collect specific datasets [7, 18, 34, 48, 63], such that the trained models can have high accuracy on the dataset if and only if they can understand the human body movement. One such example is Mimetics [63] with 713 hand-collected videos of 50 human action classes from Kinetics-400 [28] happening against irrelevant backgrounds. However, scaling up or generalizing this effort would be extremely costly due to the need for manual annotation. In contrast, our toolkit provides quantitative metrics for directly measuring the effect of background bias without the need for manual annotation.

## 3 Human-centric Analysis Toolkit

Our Human-centric Analysis Toolkit (HAT) is a general framework that can be used to measure the amount of background bias learned by a human action recognition model. HAT takes two inputs: (1) a trained human action recognition model and (2) a set of validation videos each annotated with the human action class. HAT then proceeds in three steps. First, it leverages human segmentation models to separate the human visual cues from the background visual cues in the validation videos. Second, it generates six sets of counterfactual validation videos, including Human-Only, Background-Only, and four sets of Action-Swap videos (see Figure 1 for examples). Finally, it evaluates the trained model on these counterfactual videos and returns a set of ten metrics which quantify the different effects of background bias. This methodology can expand the dataset without any need to manually collect new data, allowing deeper analysis of human action recognition in an affordable manner.

### 3.1 Separating human from background

The first step of HAT is separating the visual cues corresponding directly to the *human* from the rest of the cues in the video. This can be done using a pre-trained human segmentation model. Interestingly, in our internal experiments, we find that modern image-based segmentation models [24, 72, 69] tend to have better results than video-based segmentation models [40]. We hypothesize that this might be due to the differences in training set size. While older CNN-based image segmentation models [72, 69] suffer from low temporal consistency, missing human segments in some of the

frames, the modern transformer-based SeMask [24] appears to overcome this limitation. We use SeMask trained on ADE20K [75] in our implementation.

One thing to note is that in the current instantiation of HAT we consider any *objects* that the human is interacting to be part of the background. Thus, for example, a person performing the "drinking coffee" action would be expected to be segmented separately from the coffee mug that they are holding (which becomes part of the background). One way of partially avoiding this would be to use a human bounding box instead of a segmentation mask – however, undesirable background cues would then also be included. Different tradeoffs can be considered in future instantiations of HAT.

## 3.2 Generating counterfactual validation videos

The core of our toolkit is generating synthetic validation videos with different visual cues, which allows us to investigate the effect of the different cues on human action recognition models.

The first two sets of videos are **Background-Only** (where only the background is shown and all human cues are removed) and **Human-Only** (where only the human cues are shown). For Background-Only, we leverage the video inpainting model [29] to remove all human pixels segmented by the model of Section 3.1. In contrast to prior works [6, 21] which fill the human pixels with a frame average color value (e.g., grey), we use inpainting to generate a more realistic-looking video. For Human-Only, we instead keep only the segmented human pixels and fill in the rest with an average color. We use the *dataset's* average color rather than the *frame* average, since that can reveal a lot about the background, e.g., green for a sports field or blue for a body of water.

The other four sets of videos are more complex **Action-Swap** videos, which combine different visual cues to investigate their additive effects. We synthesize these videos by combining the segmented human figure with the background from a different video, similar to [64]. While the Background-Only and Human-Only video sets are both decidedly outside the model's training data distribution, these Action-Swap videos are arguably somewhat more realistic since they do contain a human figure against a viable background – although in an unexpected combination. Example frames are in Figure 1 and videos in supplementary material; more details on Action-Swap generation below.

### 3.2.1 Details of generating Action-Swap videos

HAT includes four different types of Action-Swap videos:

- **Random**: The background is swapped with a video from a different class.
- **Close**: The background is swapped with a video from a class with a similar background.
- **Far**: The background is swapped with a video from a class with a very different background.
- **Same**: The background is swapped with a video from the same class. This can be used as a theoretical upper bound of Action-Swap Accuracy.

To determine the appropriate classes for **Close** and **Far** Action-Swap videos, we need to determine how similar the backgrounds are across different classes. To do so, we first feed the frames from the original validation videos into a Places365 [74] trained scene classification model. For each action class, we then compute the average scene prediction vector by averaging the prediction probabilities from all frames of all videos of this class. We can then rank all the other classes according to the L1 distance in their average scene prediction vector. We consider the class to be "close" if it's among the 5 classes with the smallest L1 distance and "far" if it's among the 200 largest (of 399 classes total).

For generating an Action-Swap counterfactual video, we thus:

(1) segment the human figure from the video using [24] as if creating a Human-Only video,
(2) randomly sample a background action class, depending on the particular Action-Swap set,
(3) randomly sample a video of the class from (2),
(4) generate the Background-Only version of the video from (3),
(5) paste in the human figure from (1) onto the video from (4)

One additional challenge is that we want to ensure that sufficient human *and* background cues are present in every generated Action-Swap video. Thus, we only consider videos where all frames have human masks taking up 5-50% of the pixels; when sampling background videos in step (3) we relax

the lower bound to allow videos with few human pixels.[2] Therefore, unlike the Background-Only and Human-Only sets, the Action-Swap sets have fewer video samples than the original dataset. In Kinetics-400, we end up with 5,631 videos, whereas the original validation set has 19,877 videos. To compensate for this, we run steps (2-5) three times for each video to generate three different videos.

### 3.3  Metrics

We use the generated counterfactual validation videos from Section 3.2 to evaluate the trained human action recognition models. We measure how much of the original recognition accuracy comes from the different cues:

$$\textbf{Background-Only Ratio (BOR)} = \frac{\text{Background-Only Accuracy}}{\text{Original Accuracy}} \tag{1}$$

$$\textbf{Human-Only Ratio (HOR)} = \frac{\text{Human-Only Accuracy}}{\text{Original Accuracy}} \tag{2}$$

If a model shows high BOR, i.e., a model can get close to the original accuracy with just the background cues, we see this as "right for the wrong reason." In contrast, ideally models would have high HOR since they should be able to recognize the human action without the background cues.

Finally, for Action-Swap videos recall that each counterfactual video is generated by combining the human figure foreground from class A with the background from a different class B. We then measure the **Swap Human Accuracy (SHAcc)** as the fraction of counterfactual videos the model predicts correctly as class A, and **Swap Background Error (SBErr)** as the fraction of times the model incorrectly predicts the video as the background class B. Human action recognition models that successfully rely on human motion cues would be expected to have high SHAcc; those that are driven primarily by background cues would be expected to have high SBErr.

## 4  Analyzing Action Recognition Models

We now demonstrate the capabilities of HAT by evaluating human action recognition models trained on the popular Kinetics-400 [28] dataset. We present the results on the different types of counterfactual videos in order (Background-Only in Section 4.2, Human-Only in Section 4.3, and Action-Swap in Section 4.4), along with discussing our findings and drawing conclusions about different model design decisions that appear to have contributed to the learned background bias. HAT is not limited to Kinetics-400, and can be used on other human action recognition datasets [19, 23, 52]. Please refer to the supplementary material for the experiments on UCF101.

### 4.1  Experimental Details

We test a number of different model designs, including TSN [61], I3D [5], Non-local Neural Networks [62], R(2+1)D [54], TSM [36], SlowFast, and SlowOnly [15], CSN [58], TIN [49], TPN [68], X3D [14], OmniSource [12], TANet [39], and TimeSformer [4]. In total, we test 74 different trained models offered by the MMAction2 [8] implementation.

We extract the videos in 30 FPS with original resolution. For other pre-processing, such as resizing and temporal sampling, we follow the configuration that each model specified. We list the details of the tested models and their configuration in the supplementary material. Within the scope of the paper, we chose not to retrain any models and rely on publicly released model weights. In drawing conclusions we try to do an apples-to-apples comparison whenever possible; however, we are not able to guarantee that all hyperparameter settings are directly comparable between the different models.

For image segmentation and video inpainting, we used 20 Nvidia RTX 3090 GPUs with 20 hours of forward pass to generate synthetic videos of the full Kinetics-400 validation set. See supplementary material for examples of the synthetic videos on Kinetics-400.

### 4.2  Analysis on background-only videos

---

[2]Please see visualization examples here `https://github.com/princetonvisualai/HAT/blob/main/doc/review_discussion.md#percentage-of-synthetic-pixels`

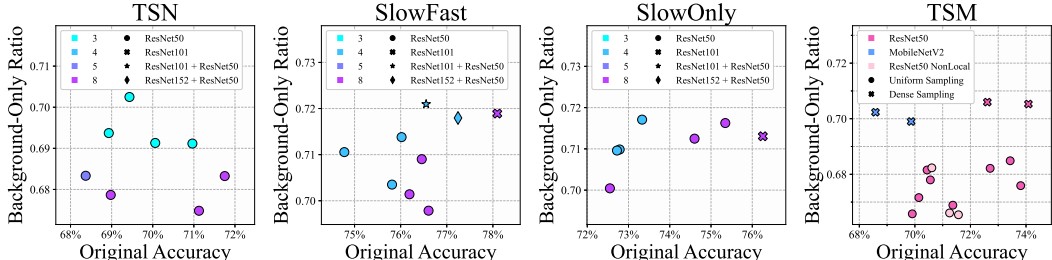

Figure 2: We plot Original Accuracy and Background-Only Ratio (BOR) of different models. **TSN, SlowFask, and SlowOnly:** Among models with similar original accuracy, models trained with fewer frames tend to show higher BOR. **TSM:** While the difference between two sampling strategies is not clear from the original accuracy, it is clear when using BOR.

Table 1: Accuracy on Background-Only Videos. When the human figure is removed, the models still tend to show high accuracy. OAcc and BAcc denote original accuracy and accuracy on Background-Only Videos, respectively. Models using additional large-scale data are tabulated separately. We only include the setting with the highest OAcc per backbone; full results of the 74 weights are in appendix.

| Model | Backbone | Pre-trained | OAcc (%) | BAcc (%) | BOR $= \frac{\text{BAcc}}{\text{OAcc}}$ |
|---|---|---|---|---|---|
| *Normal-scale dataset* | | | | | |
| TSM [36] | MNetV2 [44] | ImageNet | 69.87 | 48.84 | 0.6990 |
| R(2+1)D [54] | ResNet34 | - | 74.22 | 52.99 | 0.7140 |
| TSN [61] | ResNet50 | ImageNet | 71.75 | 49.02 | 0.6833 |
| TIN [49] | ResNet50 | TSM-Kinetics400 | 70.89 | 48.32 | 0.6816 |
| TSM [36] | ResNet50 | ImageNet | 74.09 | 52.25 | 0.7053 |
| I3D [5] | ResNet50 | ImageNet | 73.57 | 52.26 | 0.7104 |
| NL-TSM [62] | ResNet50 | ImageNet | 71.57 | 47.62 | 0.6654 |
| NL-I3D [62] | ResNet50 | ImageNet | 74.91 | 52.84 | 0.7054 |
| NL-SlowOnly [62] | ResNet50 | ImageNet | 75.78 | 53.51 | 0.7062 |
| CSN [58] | ResNet50 | - | 73.22 | 51.97 | 0.7098 |
| TPN [68] | ResNet50 | ImageNet | 76.16 | 54.40 | 0.7143 |
| SlowOnly [15] | ResNet50 | ImageNet | 75.35 | 53.97 | 0.7163 |
| SlowFast [15] | ResNet50 | - | 76.61 | 53.46 | 0.6978 |
| SlowOnly [15] | ResNet101 | - | 76.26 | 54.38 | 0.7131 |
| SlowFast [15] | ResNet101+50 | - | 76.55 | 55.19 | 0.7210 |
| SlowFast [15] | ResNet101 | - | **78.10** | 56.14 | 0.7189 |
| CSN [58] | ResNet152 | - | 77.62 | 54.33 | 0.6999 |
| SlowFast [15] | ResNet152+50 | - | 77.24 | 55.46 | 0.7179 |
| X3D [14] | X3D_S | - | 72.67 | 50.61 | 0.6964 |
| X3D [14] | X3D_M | - | 75.55 | 52.47 | 0.6944 |
| TANet [39] | TANet | ImageNet | 76.10 | 53.71 | 0.7059 |
| *Large-scale dataset* | | | | | |
| TSN [61] | ResNet50 | IG-1B [65] | 70.96 | 49.05 | 0.6912 |
| Omni-TSN [12] | ResNet50 | IG-1B [65] | 74.70 | 52.09 | 0.6973 |
| Omni-SlowOnly [12] | ResNet50 | - | 76.49 | 55.00 | 0.7190 |
| CSN [58] | ResNet50 | IG65M [16] | 79.09 | 55.83 | 0.7059 |
| Omni-SlowOnly [12] | ResNet101 | - | 80.00 | 58.05 | 0.7255 |
| CSN [58] | ResNet152 | IG65M [16] | **82.38** | 58.97 | 0.7159 |
| TimeSFormer [4] | TimeSformer | ImageNet-21K [10] | 77.97 | 53.88 | 0.6910 |

**Accuracy and Background-Only Ratio.** Table 1 tabulates the model accuracies in Background-Only Videos. For a fair comparison, we have separated the weights that use additional large-scale datasets [10, 12, 16, 26, 65]. Despite removing a human body from the video, thus removing any human action, all the models still show a strong tendency to predict the removed action. This hints at the possibility that the performance of the human action recognition models is highly dependent on the background, rather than the action itself.

Table 1 tabulates the Background-Only Ratio. It shows that on all the tested models, we see around 70% of the accuracy is coming from the non-human regions, revealing the problematic behavior,

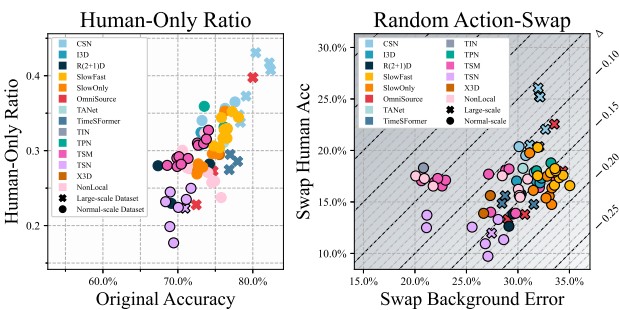

Figure 3: **Left:** Some models (e.g., CSN, OmniSource) perform consistently well on both original accuracy and HOR. However, there are some exceptions: for example, TSM (pink with black border) and TSN (violet with black border) perform similarly on original accuracy but TSM significantly outperforms TSN on HOR. **Right:** In random action-swap videos, all models are more likely to make predictions consistent with the new background (Swap Background Error) as opposed to with the human figure (Swap Human Acc).

"right for the wrong reason", is common in human action recognition. Next, we show examples of using Background-Only Ratio to analyze and improve the model design.

**Number of Frames used to Train.** The first three plots of Figure 2 visualize how the number of video frames used during the training can worsen the Background-Only Ratio. This shows that the models trained with fewer temporal frames tend to suffer more, with a lot of their accuracy coming from the background. A possible explanation is that when fewer frames are given, the model is not able to learn to understand temporal information, thus given a video with or without the human movement, the model will perform similarly, as they never learned to understand such complex human movement during training. Thus the accuracy would come from the temporally static background. While exact behavior can be different per model structure, we see this to be most severe on TSN [61] which lacks any sophisticated temporal modeling.

**Sampling Strategy.** We check if the frame sampling strategy can affect the Background-Only Ratio. The results are visualized on the last plot of Figure 2. Unlike uniform sampling, i.e., getting uniformly distributed frames, dense sampling strategy, i.e., sampling frames with a specified stride, shows higher BOR in general. We believe this is due to the dense sampling strategy having a smaller temporal window so that the model was not able to learn the body movement sufficiently. Surprisingly, the effect of the sampling strategy would have not been clear if we only used original accuracy alone (see x-axis), showing a clear benefit of using BOR for model training analysis.

### 4.3 Analysis on human-only videos

**Accuracy and Human-Only Ratio.** We plot HOR in left of Figure 3. We tabulated the evaluation results in the supplementary material. Given only the human action, all the models suffer significantly with an accuracy of around $20\%$. Despite Human-Only modification keeping the human action intact, the ratio is far lower than Background-Only Accuracy. By comparing BOR (with around $0.7$) and HOR (with around $0.3$), we quantitatively measure the well-believed problem of the current state of human action recognition, that most existing methods are all highly influenced by the background, more than the foreground human action.

Thankfully, we see a strong correlation between Human-Only Ratio and the original accuracy. This could hint that the performance improvement of the action recognition model is benefited from a better understanding of the human body, showing the important direction of where the human action recognition field needs to focus. Next, we show one example case where HOR can be used to evaluate different model structures.

**TSN vs TSM.** While the original paper on TSM [36] claims $+4\%$ accuracy improvements over TSN [61] on Kinetics-400, using different training and testing conditions, MMAction2 [61] shows that the accuracy of TSN can be achieved on par with TSM, as shown in the x-axis of left of Figure 3. However, using Human-Only Ratio as a metric, we show that TSM does indeed show superior performance over TSN when a non-human region is removed. One possible explanation is that,

Table 2: Action-Swap experiment results. We average the numbers from 3 random runs. We show standard deviation as well. See supplementary material for the full experiments.

| Model | Backbone | Pre-trained | Same SHAcc↑ | Random Swap SHAcc↑ | SBErr↓ | Close SHAcc↑ | SBErr↓ | Far SHAcc↑ | SBErr↓ |
|---|---|---|---|---|---|---|---|---|---|
| *Normal-scale dataset* | | | | | | | | | |
| TSM [36] | MNetV2 | ImgNet | $62.2_{\pm.3}$ | $13.9_{\pm.1}$ | $29.8_{\pm.2}$ | $24.4_{\pm.3}$ | $26.4_{\pm.4}$ | $11.2_{\pm.1}$ | $35.5_{\pm.2}$ |
| R(2+1)D [54] | Res34 | - | $64.5_{\pm.3}$ | $15.8_{\pm.3}$ | $30.3_{\pm.5}$ | $26.6_{\pm.3}$ | $27.1_{\pm.4}$ | $13.0_{\pm.1}$ | $35.6_{\pm.1}$ |
| TSN [61] | Res50 | ImgNet | $60.2_{\pm.2}$ | $13.3_{\pm.1}$ | $28.1_{\pm.2}$ | $23.4_{\pm.2}$ | $26.7_{\pm.3}$ | $11.9_{\pm.1}$ | $32.7_{\pm.2}$ |
| TIN [49] | Res50 | Kin400 | $58.6_{\pm.1}$ | $18.3_{\pm.2}$ | $\mathbf{20.8}_{\pm.1}$ | $27.1_{\pm.2}$ | $\mathbf{21.0}_{\pm.2}$ | $16.6_{\pm.1}$ | $\mathbf{23.5}_{\pm.3}$ |
| TSM [36] | Res50 | ImgNet | $66.6_{\pm.4}$ | $17.2_{\pm.5}$ | $33.7_{\pm.5}$ | $27.8_{\pm.1}$ | $29.2_{\pm.2}$ | $14.3_{\pm.3}$ | $40.4_{\pm.2}$ |
| I3D [5] | Res50 | ImgNet | $64.9_{\pm.4}$ | $17.0_{\pm.2}$ | $29.9_{\pm.1}$ | $27.4_{\pm.3}$ | $26.6_{\pm.5}$ | $14.8_{\pm.5}$ | $34.8_{\pm.5}$ |
| NL-TSM [62] | Res50 | ImgNet | $58.6_{\pm.4}$ | $16.5_{\pm.2}$ | $21.8_{\pm.2}$ | $25.9_{\pm.6}$ | $21.7_{\pm.2}$ | $15.0_{\pm.3}$ | $25.0_{\pm.1}$ |
| NL-I3D [62] | Res50 | ImgNet | $64.9_{\pm.4}$ | $16.2_{\pm.2}$ | $30.0_{\pm.4}$ | $27.0_{\pm.2}$ | $26.6_{\pm.1}$ | $13.4_{\pm.3}$ | $35.6_{\pm.4}$ |
| NL-SlowOnly [62] | Res50 | ImgNet | $63.8_{\pm.1}$ | $17.5_{\pm.2}$ | $28.5_{\pm.5}$ | $27.0_{\pm.2}$ | $25.6_{\pm.3}$ | $14.8_{\pm.5}$ | $34.0_{\pm.4}$ |
| CSN [58] | Res50 | - | $65.9_{\pm.3}$ | $17.9_{\pm.2}$ | $31.6_{\pm.2}$ | $28.2_{\pm.2}$ | $27.6_{\pm.5}$ | $15.2_{\pm.2}$ | $37.1_{\pm.5}$ |
| TPN [68] | Res50 | ImgNet | $69.3_{\pm.2}$ | $18.8_{\pm.2}$ | $33.2_{\pm.5}$ | $29.0_{\pm.3}$ | $29.0_{\pm.6}$ | $15.8_{\pm.2}$ | $38.9_{\pm.4}$ |
| SlowOnly [15] | Res50 | ImgNet | $68.2_{\pm.1}$ | $17.5_{\pm.2}$ | $32.8_{\pm.5}$ | $28.1_{\pm.4}$ | $28.7_{\pm.2}$ | $14.8_{\pm.3}$ | $38.8_{\pm.4}$ |
| SlowFast [15] | Res50 | - | $68.4_{\pm.3}$ | $18.0_{\pm.3}$ | $33.7_{\pm.6}$ | $28.8_{\pm.2}$ | $29.7_{\pm.5}$ | $15.0_{\pm.2}$ | $40.0_{\pm.2}$ |
| SlowOnly [15] | Res101 | - | $69.4_{\pm.4}$ | $19.8_{\pm.3}$ | $31.1_{\pm.6}$ | $\mathbf{31.0}_{\pm.1}$ | $28.1_{\pm.3}$ | $17.0_{\pm.2}$ | $37.0_{\pm.4}$ |
| SlowFast [15] | Res101+50 | - | $67.9_{\pm.2}$ | $17.5_{\pm.3}$ | $31.9_{\pm.4}$ | $28.4_{\pm.3}$ | $29.0_{\pm.2}$ | $15.1_{\pm.1}$ | $37.7_{\pm.6}$ |
| SlowFast [15] | Res101 | - | $\mathbf{69.6}_{\pm.3}$ | $18.2_{\pm.3}$ | $33.6_{\pm.6}$ | $29.2_{\pm.3}$ | $29.4_{\pm.3}$ | $15.4_{\pm.1}$ | $40.0_{\pm.5}$ |
| CSN [58] | Res152 | - | $67.8_{\pm.4}$ | $\mathbf{20.4}_{\pm.5}$ | $30.1_{\pm.3}$ | $30.8_{\pm.2}$ | $26.3_{\pm.3}$ | $\mathbf{17.6}_{\pm.3}$ | $35.2_{\pm.0}$ |
| SlowFast [15] | Res152+50 | - | $69.3_{\pm.5}$ | $20.3_{\pm.6}$ | $31.9_{\pm.7}$ | $31.0_{\pm.1}$ | $28.5_{\pm.3}$ | $17.5_{\pm.2}$ | $36.9_{\pm.2}$ |
| X3D [14] | X3D_S | - | $60.8_{\pm.3}$ | $13.9_{\pm.3}$ | $26.7_{\pm.7}$ | $24.2_{\pm.2}$ | $24.7_{\pm.3}$ | $11.0_{\pm.1}$ | $32.0_{\pm.3}$ |
| X3D [14] | X3D_M | - | $64.3_{\pm.3}$ | $15.6_{\pm.2}$ | $27.3_{\pm.1}$ | $26.5_{\pm.4}$ | $25.5_{\pm.1}$ | $12.8_{\pm.0}$ | $32.8_{\pm.6}$ |
| TANet [39] | TANet | ImgNet | $67.1_{\pm.3}$ | $18.3_{\pm.3}$ | $30.5_{\pm.4}$ | $28.5_{\pm.2}$ | $27.0_{\pm.3}$ | $15.5_{\pm.1}$ | $36.6_{\pm.4}$ |
| *Large-scale dataset* | | | | | | | | | |
| TSN [61] | Res50 | IG-1B [65] | $57.7_{\pm.5}$ | $12.0_{\pm.3}$ | $\mathbf{27.4}_{\pm.3}$ | $21.4_{\pm.3}$ | $\mathbf{25.7}_{\pm.1}$ | $10.1_{\pm.3}$ | $\mathbf{32.1}_{\pm.2}$ |
| Omni-TSN [12] | Res50 | IG-1B [65] | $63.9_{\pm.6}$ | $13.8_{\pm.4}$ | $30.7_{\pm.1}$ | $24.4_{\pm.1}$ | $27.9_{\pm.2}$ | $11.8_{\pm.2}$ | $36.8_{\pm.6}$ |
| Omni-Slow [12] | Res50 | - | $69.5_{\pm.3}$ | $18.0_{\pm.6}$ | $34.4_{\pm.5}$ | $29.1_{\pm.2}$ | $29.8_{\pm.2}$ | $15.0_{\pm.2}$ | $40.8_{\pm.2}$ |
| CSN [58] | Res50 | IG65M [16] | $70.4_{\pm.3}$ | $22.1_{\pm.5}$ | $32.7_{\pm.2}$ | $32.4_{\pm.4}$ | $28.9_{\pm.4}$ | $18.8_{\pm.1}$ | $38.7_{\pm.2}$ |
| TSFormer [4] | TSformer | Img21K [10] | $65.3_{\pm.3}$ | $15.6_{\pm.3}$ | $28.8_{\pm.1}$ | $25.8_{\pm.1}$ | $27.4_{\pm.3}$ | $13.0_{\pm.3}$ | $33.2_{\pm.5}$ |
| Omni-Slow [12] | Res101 | - | $\mathbf{73.3}_{\pm.4}$ | $22.6_{\pm.2}$ | $33.5_{\pm.4}$ | $33.4_{\pm.5}$ | $30.1_{\pm.5}$ | $19.4_{\pm.2}$ | $39.2_{\pm.3}$ |
| CSN [58] | Res152 | IG65M [16] | $72.9_{\pm.1}$ | $\mathbf{25.2}_{\pm.4}$ | $32.2_{\pm.5}$ | $\mathbf{35.6}_{\pm.3}$ | $28.4_{\pm.6}$ | $\mathbf{22.1}_{\pm.3}$ | $38.0_{\pm.3}$ |

as TSM design makes use of temporal difference, e.g., human body movement, it can capture the information of the human body better, as TSN cannot distinguish between human and background using its basic temporal modeling design.

### 4.4 Analysis on Action-Swap videos

**Accuracy on Action-Swap.** Table 2 and right of Figure 3 details the performance of different models over the Action-Swap Videos. It shows that when we randomly swap background with other videos, all the models lean towards predicting the class of the background, rather than the foreground human action. Swapping between classes that are similar/different shows a gain/drop in SHAcc, showing that the output of a human action recognition model is largely dependent on the background.

**Original Accuracy vs. Action-Swap Accuracy.** Among models using normal-scale datasets, SlowFast-Res101 [15] shows the best accuracy when the background is relevant to the foreground action, on both original Kinetics accuracy (See Tab. 1) and Same Swap (See Tab. 2). However, given counterfactual videos that have irrelevant backgrounds, their performance drops to 18%, while the model falsely predicts 34 percent of the validation videos as their background class, one of the highest among the models we have tested. Such low performance on human action could be due to its reliance on the background, as models with better Random Swap SHAcc (CSN-Res152, SlowFast-Res152+50, etc.) show fewer background errors. Such experiment shows that models showing good accuracy in original Kinetics-400, might not be a good human action recognition model, due to their reliance on the background.

Table 3: Performance comparison when using a Non-local module [62]. NL-EG, NL-G, and NL-Dot denote Non-local method using embedded Gaussian, Gaussian, and dot product, respectively. Numbers are bolded when the Non-local module improves the metric.

| Model | frames | OAcc↑ | SHAcc↑ | SBErr↓ |
|---|---|---|---|---|
| TSM | 8 | 72.89 | 16.55 | 22.64 |
| TSM + NL-EG | 8 | $\mathbf{74.06}_{(+1.18)}$ | $16.54_{(-0.01)}$ | $\mathbf{21.77}_{(-0.86)}$ |
| TSM + NL-G | 8 | $72.61_{(-0.27)}$ | $\mathbf{17.52}_{(+0.98)}$ | $\mathbf{20.07}_{(-2.56)}$ |
| TSM + NL-Dot | 8 | $\mathbf{73.52}_{(+0.63)}$ | $\mathbf{17.27}_{(+0.73)}$ | $\mathbf{20.91}_{(-1.72)}$ |
| I3D | 32 | 75.33 | 17.05 | 31.78 |
| I3D + NL-EG | 32 | $\mathbf{76.90}_{(+1.58)}$ | $16.23_{(-0.83)}$ | $\mathbf{30.04}_{(-1.73)}$ |
| I3D + NL-G | 32 | $\mathbf{75.96}_{(+0.63)}$ | $\mathbf{17.22}_{(+0.17)}$ | $\mathbf{30.94}_{(-0.83)}$ |
| I3D + NL-Dot | 32 | $\mathbf{76.17}_{(+0.84)}$ | $15.63_{(-1.43)}$ | $\mathbf{30.14}_{(-1.64)}$ |
| SlowOnly | 4 | 75.28 | 14.75 | 33.27 |
| SlowOnly + NL-EG | 4 | $\mathbf{76.10}_{(+0.82)}$ | $\mathbf{15.46}_{(+0.70)}$ | $\mathbf{30.21}_{(-3.07)}$ |
| SlowOnly | 8 | 75.18 | 16.12 | 31.49 |
| SlowOnly + NL-EG | 8 | $\mathbf{77.74}_{(+2.56)}$ | $\mathbf{17.54}_{(+1.42)}$ | $\mathbf{28.48}_{(-3.01)}$ |

Table 4: Performance when using a large-scale dataset. We compare the same settings except for the initial weight. Numbers are bolded when the large-scale dataset improves the metric.

| Model | Backbone | Pre-trained | OAcc↑ | SHAcc↑ | SBErr↓ |
|---|---|---|---|---|---|
| TSN [61] | ResNet50 | ImageNet | 72.55 | $11.34_{\pm0.15}$ | $28.62_{\pm0.16}$ |
| TSN [61] | ResNet50 | IG-1B | **73.39** | $\mathbf{11.96}_{\pm0.35}$ | $\mathbf{27.45}_{\pm0.28}$ |
| ir-CSN [58] | ResNet50 | None | 75.51 | $17.88_{\pm0.18}$ | $31.58_{\pm0.17}$ |
| ir-CSN [58] | ResNet50 | IG65M | **81.46** | $\mathbf{22.05}_{\pm0.49}$ | $32.68_{\pm0.17}$ |
| ir-CSN [58] | ResNet152 | None | 78.08 | $19.51_{\pm0.11}$ | $30.76_{\pm0.23}$ |
| ir-CSN [58] | ResNet152 | Sports1M | **78.98** | $\mathbf{20.52}_{\pm0.21}$ | $31.14_{\pm0.51}$ |
| ir-CSN [58] | ResNet152 | IG65M | **83.17** | $\mathbf{25.25}_{\pm0.36}$ | $32.07_{\pm0.39}$ |
| ip-CSN [58] | ResNet152 | None | 79.26 | $20.37_{\pm0.50}$ | $30.11_{\pm0.34}$ |
| ip-CSN [58] | ResNet152 | Sports1M | **79.38** | $20.37_{\pm0.36}$ | $32.06_{\pm0.31}$ |
| ip-CSN [58] | ResNet152 | IG65M | **83.92** | $\mathbf{25.19}_{\pm0.41}$ | $32.16_{\pm0.46}$ |

**Use of Non-local Module.** To demonstrate the evaluation of a model design using Action-Swap, we select Non-local [62] module as an example. Table 3 tabulates the evaluation results on Random Swap. We see that the Non-local module not only improves the original accuracy, but also drops the background error on all the tested models, showing reduced background bias. However, Non-local module do not always improve the focus on the human body, as for I3D [5] models, we see that SHAcc tends to drop.

**Use of Large-scale Dataset for Pre-training.** Table 4 tabulates the performance of models where we compare trained weight with/without additional large-scale pre-training. It shows that in all the cases, using a large-scale dataset improves the original accuracy and Random Swap Human Accuracy. However, as CSN shows an increase in the Background Error, this does not necessarily mean that the model is being better at recognizing the human. We expect the model is recognizing the image feature better when pre-trained with large-scale dataset, regardless of the scene or the person.

**Comparison with Existing Methods.** We compare Mimetics [63] dataset accuracy and Random Action-Swap SHAcc by evaluating different models. Given that Mimetics dataset contains non-synthetic counterfactual videos where the action is performed on unrelated backgrounds, i.e., non-synthetic version of Action-Swap, we expect SHAcc to show correlations with Mimetics accuracy. The (a) of Figure 4 visualizes the comparisons between Mimetics and SHAcc. As expected, we see that there is a strong correlation between Mimetics accuracy and SHAcc. This shows that our synthetic counterfactual dataset can bring similar conclusions as using non-synthetic ones.

Moreover, we compare our metric with the pointing game, a popular methodology [21, 70] of converting GradCAM into a quantitative metric. To convert, one needs to generate an activation map and count whether the highest activation point falls into the target segmentation or not. Here, we choose TSN model and use the activation map of the second last ReLU of the penultimate layer and check if the highest activation point falls into the human segmentation. In (b) of Figure 4 we plot pointing game evaluation and random Action-Swap SHAcc of different classes. (c-d) of Figure 4 plot each of SHAcc and pointing game with regards to human segmentation size over the image size.

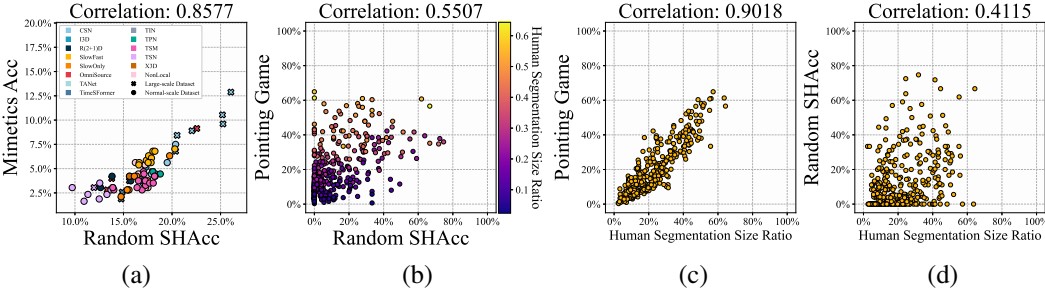

Figure 4: **(a):** SHAcc and Mimetics accuracy of different trained models. Our SHAcc using synthetic videos is strongly correlated with the results on the manually-collected Mimetics videos. **(b):** In contrast, our SHAcc metric is *not* strongly correlated with the pointing game. This is likely because the pointing game is strongly correlated with the human size (c), whereas our metric is not (d).

We see that the pointing game seems to be heavily affected by the size of the human figure. The pointing game evaluation have a large correlation coefficient of $0.9018$, thus favoring the videos with large human segmentation. Such correlation is less visible in our SHAcc metric that has a correlation coefficient of $0.4115$. This result shows that our metric may be a more suitable metric for evaluating background bias of human action recognition models.

## 5    Conclusion

We introduce a general framework for human-centric analysis for human action recognition models. We test Human-centric Analysis Toolkit on the Kinetics-400 dataset and evaluate the generated dataset on a number of existing action recognition models.

Through extensive experiments over 74 trained models, we find that all the models we tested have stronger background bias. However, we found that the background bias can be mitigated when more frames are fed during the training, the temporal stride between frames is increased, and temporal/spacial modeling is improved using Non-local module. Moreover, we see that the original accuracy do not fully represent the human understanding as the accuracy cannot differentiate TSN and TSM, large-scale dataset and Non-local module improves original accuracy but not necessarily SHAcc. Lastly, we show that using our generated dataset can bring similar conclusions as using a non-synthetic counterfactual dataset.

From our findings, we suggest the future researchers to (1) not rely on the accuracy as the only metric, as original accuracy do not fully represent the performance of the model based on the human action; (2) carefully select the temporal hyper-parameters, as temporal parameters can improve/worsen the background bias of human action recognition models; and (3) use HAT toolkit to see if the model design (e.g., as Non-local) can improve your model on accuracy and reduce the background bias. We hope that this tool can be adopted by future researchers for a better human-centric analysis of human action recognition models.

## 6    Discussion

**Limitation** As we use an off-the-shelf image semantic segmentation model and a video inpainting model, the quality of the synthetic dataset is limited by the performance of the aforementioned models. **Ethical Concerns** Our tool requires the use of image segmentation and inpainting tool to generate a dataset, requiring computation cost for the initial setup. However, as human-centric analysis using our tool does not require any new training, we believe our tool is more environmentally friendly than the existing methods. Moreover, as our tool is automated, human labor for data collection is not required. Also, as we generate a dataset from an existing dataset, we show fewer concerns about privacy issues when a new video dataset is generated.
**License** MMAction2 [8] and SeMask [24] follow Apache License 2.0. We used author-released code for Deep Video Inpainting [29] which did not specify any license. Kinetics-400 annotation data is licensed under a Creative Commons Attribution 4.0 International License, but some of the video sources do not specify any license. Please refer to the individual licenses when using our released code.

**Acknowledgements.** We are grateful for the support from the National Science Foundation under Grant No. 2112562, Microsoft, Princeton SEAS Project X Innovation Fund, and Princeton First Year Ph.D. Fellowship to JC. We thank Angelina Wang for the helpful feedback.

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
