# OpenReview forum: "Enabling Detailed Action Recognition Evaluation Through Video Dataset Augmentation"
_NeurIPS.cc/2022/Track/Datasets_and_Benchmarks — NeurIPS 2022 Datasets and Benchmarks _

### Official Review · Reviewer_dwoF · 2022-07-19

**Rating:** 7
**Confidence:** 4
**Clarity:** The paper is well-written.

**Strengths:**

i) Important and relevant problem:
We all know that there's a big difference between just guessing and actually seeing it with concrete experimental results.
From this perspective, the paper's empirical results clarify the CV researcher's vague guess, which stimulates further research on the problem.
I really appreciate the author's effort in conducting such valuable experiments.

ii) Convincing experimental settings:
I think Only background, Only Human-action, Swap Action setting is reasonable, and the presented empirical results are interesting.

iii) Extensive experiments on different models:
Exhaustive experiments are conducted and the results are fairly reported.

Overall, I think that the experiments are convincing, and have a meaningful impact on the CV community.

**Weaknesses:**

i) Straightforward method and expectable empirical results:
Utilizing off-the-shelf segmentation/inpainting models is quite straightforward, lacking technical novelty.
Besides, the empirical results in the paper double-check the well-known prior, rather than suggesting novel insights.
As the paper is not proposing a new dataset, I think at least some level of technical novelty or some new findings is needed.

ii) Suboptimal behavior of the off-the-shelf models:
Using off-the-shelf models has an inherent limitation due to the suboptimal performance of the adopted off-the-shelf models.
It is explicitly mentioned in the paper as well and can be clearly seen in the videos in the supplementary materials.
Thus, I think that the suboptimal behavior of the off-the-shelf model may affect the experimental results in some sense, but we do not exactly know ``how much'' it affected the results.

iii) camera--ready release claim:
The authors noted that "We plan to offer the official implementation of the toolkit by the release of the camera-ready version of the paper." in their supplementary materials.
As the track is about datasets and benchmarks, I think it should be open at the review period, not at the camera-ready period.
I think it is the most critical issue in this submission.

I really appreciate the authors' effort in conducting such thorough experiments, and think the empirical results are valuable for the CV community. (Much higher than the acceptance threshold) However, unless the toolkit is provided with sufficient documentation, I cannot recommend acceptance of the paper.


**Additional Feedback:**

I'll be happy to raise my rating if the authors provide access/good documentation of the toolkit.

**Correctness:**

Reasonable experimental settings, but as it uses off-the-shelf ML models to preprocess video, the empirical results here might not exactly match with the results on real videos.

**Documentation:**

At this point, I cannot evaluate the documentation quality of the toolkit.


**Relation To Prior Work:**

Related works are clearly presented.

**Summary And Contributions:**

Many CV researchers know (or guess?) that the action recognition models tend to see the ``background scene'' of the video, rather than the human action itself.
However, a thorough inspection of the issue is hard to be conducted, because of the lack of a valid dataset.
The authors detour this dataset problem by adopting off-the-shelf inpainting/segmentation models.
Extensive experiments are conducted, and the results are reported thoroughly.

---

> ### Author Response · Authors · 2022-08-24
> **Official Comment**
>
> Dear Reviewer dwoF,
>
> Thank you for your thoughtful comments and suggestions. Apologies for the delay – we were doing a detailed revision to incorporate all the feedback from the reviewers, and wanted to make sure that is finalized before responding.
>
> The toolkit is now available at https://github.com/princetonvisualai/HAT. The updated paper is uploaded.
>
> Here are answers to some of the other questions that you have mentioned. We hope you will reconsider your final score (since we are still within the allotted rebuttal window).
>
> - Re: Using existing methods lacks technical novelty. The results check well-known prior, without novel insights.
>
> We politely disagree that the paper lacks novelty, but can see how the original writing might make you think so. We rewrote Section 3 to better highlight the contributions of our tool. While it leverages existing segmentation and inpainting tools, our novelty lies in using these existing methods to design the first toolkit that quantitatively evaluates the effect of background on human action recognition models. We also rewrote the introduction to clarify that while the issue of background bias is well-known (first paragraph), it’s really quantifying this bias that is novel (second paragraph; see also the “Human-centric Analysis” subsection of Section 2). Finally, we include a number of takeaways in the introduction (lines 45-56). To the best of our knowledge, we are the first to empirically verify the first takeaway “all 74 [human action recognition models] exhibit strong background bias” and the first to observe the second and third takeaways: “models trained with fewer frames per video appear to be more prone to background bias” and “models trained with dense temporal sampling around a single timestep appear to be more prone to background bias compared to models trained with uniform sampling throughout the video.”
>
> - Re: Using off-the-shelf models has suboptimal performance.
>
> Although the off-the-shelf inpainting and segmentation models are not perfect, as some reviewers agreed, qualitatively they look promising (check the attached videos in the supplementary material). Further, as these models improve, HAT can seamlessly be updated with the newer versions.
>
> Using synthetic data to understand the model behavior is widely used in the image domain. Notably, [1,2,3] used a pasted object in an image to study the object recognition/detection model, and [4,5,6] use a grid-like image on image recognition models. Following these works, we propose using synthetic video to evaluate trained models.
>
> Nevertheless, your point is well-taken. One result which might partially mitigate this concern is noting that the human action recognition accuracy on Same Action-Swap videos (where the human figure segmented from one video is pasted onto the background from a different video but from the same action class) is not dramatically dissimilar from the original human action recognition accuracy (evaluated on real videos). For example, the SlowFast ResNet101 model has Same Action-Swap accuracy of 69.6% (Table 2, “Same” column) compared to the original accuracy of 78.1% (Table 1, “OAcc” column).
>
> [1] The elephant in the room, Amir Rosenfeld, et. al., arXiv, 2018
> [2]  Strike (With) a Pose: Neural Networks Are Easily Fooled by Strange Poses of Familiar Objects, Michael A. Alcorn, et. al., CVPR, 2019
> [3] Noise or Signal: The Role of Image Backgrounds in Object Recognition, Kai Yuanqing Xiao, et. al., ICLR 2021
> [4] Convolutional Dynamic Alignment Networks for Interpretable Classifications, Moritz B ̈ohle, et. al., CVPR 2021
> [5] Towards Better Understanding Attribution Methods, Sukrut Rao, et. al., CVPR 2022
> [6] B-cos Networks: Alignment is All We Need for Interpretability, Moritz B ̈ohle, et. al., CVPR 2022
>
>
>
>
> - Re: “how much” suboptimal tools affected the results.
>
> We have included an additional ablation study in the supplementary material where we compare the current inpainting model and a naive method of impainting (Section 1.2 in the revised supplementary materials). We hope that this experiment might give a glimpse of an idea of how the quality of the method can affect the experimental results.

---

> > ### Comment · Reviewer_dwoF · 2022-08-24
> > **Thank you for providing good response!**
> >
> > Most of my concerns are addressed well (especially the toolkit release concern). I updated my rating to 7.

---

### Official Review · Reviewer_5jrL · 2022-07-25
**Review of "Enabling Detailed Action Recognition Evaluation Through Video Dataset Augmentation"**

**Rating:** 7
**Confidence:** 3
**Correctness:** The paper's claims are well-motivated…

**Strengths:**

- This paper provides a helpful tool for analyzing a specific, known limitation of action recognition models -- background dependence. It uses this tool to produce a dataset in which humans are separated from background scenes and placed in new background scenes, to analyze the background-dependence of each model. It evaluates 74 different action recognition models on this dataset to understand how and if these models depend on scene background during classification.

- The paper's measurements are thorough, and the evaluation metrics succinctly capture the problem at hand, i.e. measuring how models perform when they see humans only vs. background only.



**Weaknesses:**

- Overall comment: While I appreciated the thorough evaluation of the background-scene dependence problem in this paper, the paper could be more thoughtful in its motivation and presentation of results. The tables and figures can be hard to read, and the authors do not explain well why they choose to present certain results but move other results to the Appendix. Finally, the authors provide few insights for addressing the problem highlighted in their work.

Specific comments:
- Results are reported in a confusing manner. For example, the paper claims to have evaluated on 74 different models (c.f. 4.1), but results on far fewer than 74 models are reported in the main paper body, with no explanation of why particular results are reported in main body vs. the supplementary. An example is Table 1, which reports results on 12 models and does not even cover all of the model designs listed in 4.1. Similarly, Figure 2 only presents results on 4 models -- why are these models chosen? This presentation should be better motivated and more systematic.

- Table 1 -- many results are said to be averaged across different settings. What settings? Why are no error bars reported?

- Figure 2:
1. Color scheme must be more distinct. It is impossible to distinguish between "3" and "4" in the leftmost 3 figures.
How can you be sure that the number of frames used for training is the cause of the higher BOR in the leftmost Fig 2 graph, rather than the model structure itself? An ablation of model structure vs. different frame rates is never presented, so it's impossible to disentangle these two factors.
2. Error bars are not provided. Given the tight range of y-axis points in all four graphs (e.g. range is < 3 percentage points), it seems likely that the results may not be statistically different.

- Figure 3:
1. Bottom right figure: should x-axis be "Swap background err" rather than "Swap background acc"?
2. Color for TSN changes between the left and right figures, which was confusing.

- Table 4:
1. what is the "background category"? Is this defined as the label of the action originally performed in this scene, before the human was removed?
2. How to read this table? Are the elements in each line separate or related somehow (e.g. a belly dancer placed in "exercising arm" setting?)

- The paper analyzes a specific problem (background-dependence in action recognition models), but provides no tangible suggestions for future work beyond a few comments in Section 4. The research community would certainly benefit from hearing more future work suggestions towards improving this problem based on the analysis in this paper.

**Additional Feedback:**

- It seems like low-hanging fruit would be to run HAT on an action recognition TRAINING dataset to produce a dataset of human-only videos and observe if background-dependence decreases in this new model. Why did the authors not perform this analysis?

- Why is it important that action recognition models detangle human action from the background scene? It seems like context would be important in distinguishing say, holding a baseball bat vs. holding a rolling pin. It is not clear from the paper why this separation is needed, rather it is just implied that it is important. This should be better motivated in the abstract/introduction.

- Section 3: could use more narrative text up front introducing WHY you create the three types of videos. The reader must intuit the motivation as being because "background-only" and "human-only" enable the creation of "swap".

**Clarity:**

- Overall, the paper is well-laid out but could use more narrative text motivating specific actions and evaluations. For example, Section 4 drops the reader right into experimental details and evaluation without first explaining why the evaluation is ordered the way it is or motivating the particular experiments run.

- The story of the paper is a bit confusing. The intro/abstract focus a lot on the need to reduce the cost of video dataset development, but the rest of the paper then focuses on/evaluates the effect of background scene on action recognition model performance via augmentations of an existing dataset. Based on the intro/abstract, I thought the paper was going to propose ways of creating new (training) datasets from existing video datasets, but instead it focuses on developing a specific validation set. Just a note that I found this hard to follow.

- Some small grammar errors (subject/verb alignment, etc), especially in intro (lines 23-24). "Tabel" --> "Table", line 214

**Documentation:**

Sufficient documentation is provided.

**Ethics:**

No ethical concerns.

**Relation To Prior Work:**

- Prior work is well-covered.

- Question: why can't you analyze the weights of a model trained on a specific dataset (para 3 of "Human-centric analysis" in Sec 2)? This assertion is made but not backed up. GradCAM (or related techniques) seem like they would still be appropriate for analyzing these models.

**Summary And Contributions:**

This paper introduces a new metric for studying the impact of background scene on the performance of action recognition models and provides a toolkit for computing it. The tool uses existing human segmentation and in-painting models to place humans in different background scenes to evaluate the impact of pose vs. background on model performance. The paper then uses this tool to evaluate the background-dependence of 74 existing action recognition models.

---

> ### Author Response · Authors · 2022-08-24
> **Official Comment**
>
> Dear Reviewer 5jrL,
>
> We thank you for the detailed comments.
>
> We understand your concerns regarding the story of the paper;  point very well taken. We completely revised the introduction to incorporate a more complete motivation for our work (beyond just the cost of creating new video datasets), as well as to explicitly include some key takeaways from our experimental results. We also completely re-organized Section 3 to more clearly present our method. Finally, we smoothed the lead-in to Section 4, and added more context and takeaways throughout. Please let us know if you have any additional thoughts or suggestions. We know there’s not much time left but we would very much appreciate your feedback.
>
> Here are answers to the other questions and comments you have listed.
>
> **Weaknesses:**
>
> - Re: Choice of presented models.
>
> We have revised the paper so that the table shows results from all the model architectures and the backbones. Among trained weights that share the same model architectures and the backbone, we select the setting where the original accuracy is the highest (we clarify this in the caption of Table 1).  We see such settings to be the most representative setting within each backbone.
>
> Figure 2 includes only 4 model architectures as some architectures do not have enough released model weights to do any comparison (clarified in lines 205-207). We have only included the architectures where we can do a fair comparison of a specific setting.
>
>
> - Re: Averaged settings in table 1.
>
> In the original version, due to the page restrictions, we have averaged the values that share a common architecture and backbone. After the reviews, we have decided that averaging between different settings can be misleading, and are now showing the results of a single setting (the one with the highest original accuracy) per architecture and backbone (added in the caption of Table 1.).
>
>
> - Re: Color scheme of figure 2.
>
> We changed the color scheme. Thank you for the suggestion.
>
>
> - Re: the number of frames used for training is the cause of the higher BOR in the leftmost Fig 2 graph.
>
> In the leftmost figure, we visualize TSN model architecture only, where they all share the ResNet50 backbone. As they all share the same model structure, we can see the effect resulting from the number of frames.
> As each of the dots represent models training in different settings (not only number of frames, but also resolution, stride, etc), through Figure 2, we only make conclusions when we have enough dots to compare between settings.
>
>
> - Re: Error bars in Figure 2.
>
> Each dot represents the accuracy of a single trained weight, where the accuracies are fixed values (in the caption of Figure 2). So the error bar cannot be provided. We only made a conclusion from the cases where we have enough dots to make comparisons.
>
>
> - Re: In Figure 3, rename the x-axis label and the color should be consistent.
>
> We have fixed the label. Thank you.
>
>
> - Re: In Table 4, what is the background category?
>
> For the action swapped video, a background category is a class label of the new background. Vise versa, the action category is the class label of the action in the swapped video.
>
>
> - Re: How to read Table 4
>
> The table shows the Average SHAcc and SBErr per class category. For example, Capoeira on the left means that among all the swapped videos where the action is Capoeira, we calculate SHAcc and SBErr. Vice versa, Exercising arm on the right means that the values are from all the action swapped videos having an exercising arm background. We include a better explanation (lines 64-68 in the revised submission of supplementary).
>
> - Re: Lack of recommendation for future model design.
>
> Thank you for the suggestion. We summarize the findings in lines 291-297, and we have provided the recommendation for future model design in the conclusion (added in lines 298-304).
>
> “From our findings, we suggest the future researchers to (1) not rely on the accuracy as the only metric, as original accuracy do not fully represent the performance of the model based on the human action; (2) carefully select the temporal hyper-parameters, as temporal parameters can improve/worsen the background bias of human action recognition models; and (3) use HAT toolkit to see if the model design (e.g., as Non-local) can improve your model on accuracy and reduce the background bias. We hope that this tool can be adopted by future researchers for a better human-centric analysis of human action recognition models.”
>
>
> **Clarity:**
>
> - Re: Grammar errors.
>
> We have revised the grammar issues. Thank you.
>
>
> **Relation To Prior Work:**
>
> - Re: Why can't you analyze the weights of a model trained on a specific dataset?
>
>
> Yup, that was confusing; we rewrote the entire paragraph, please see lines 83-101.

---

> > ### Author Response · Authors · 2022-08-24
> > **Official Comment**
> >
> >
> >
> > **Additional Feedback:**
> >
> > - Re: Use HAT for training.
> >
> > Using a modified dataset for training is a possible future direction that we want to work on. However, currently, we are focusing just on building the evaluation toolkit. To the best of our knowledge, ours is the first such toolkit that allows quantitative measurements of human action recognition models’ sensitivity to background cues without the need for additional data collection.
> >
> >
> > - Re: Talk about detangling human action from the background scene in the abstract/introduction.
> >
> >
> > We have rewritten the introduction and now it contains rich discussions related to the need of detangling human action (added in lines 16-22) and the difficulties of interpreting machine learning models (added in lines 23-29). Moreover, detangling human action and out-of-context recognition are included as additional related works in the supplementary material (Section 2 of supplementary material).
> >
> >
> > - Re: Narrative text on why you created three types of videos.
> >
> > We include reasoning on why we created these types of videos in the methods (e.g., lines 130-131, lines 140-141 in the revised submission).
> >
> > “The core of our toolkit is generating synthetic validation videos with different visual cues, which allows us to investigate the effect of the different cues on human action recognition models.”
> > “The other four sets of videos are more complex Action-Swap videos, which combine different visual cues to investigate their additive effects.”

---

> > > ### Comment · Reviewer_5jrL · 2022-08-24
> > > **Response to Revision**
> > >
> > > Thanks for making these changes. I've updated my rating.

---

### Official Review · Reviewer_2PNT · 2022-07-26
**Role of background in action recognition**

**Rating:** 7
**Confidence:** 5

**Strengths:**


The discussion about the role of background vs. real action recognition and which visual cues play which role in action classification has been around for a long time.
The proposed method and the respective data could help to assess this question for existing and future models in a consistent and reproducible manner.

With that respect, the proposed dataset could have a significant impact on the deeper analysis of action classification models and architectures, mainly doing for action recognition what ObjectNet did for ImageNet.

Furthermore, the availability of the used software could allow expanding those tests also to other datasets.

Note that the assessment of strengths is based on the assumption that the newly generated data and software will be made available to serve as a new benchmark for the community.

**Weaknesses:**

There are some questions open for me that could have a direct impact on the quality and usefulness of the proposed work:

- Do the new derivates (human-only, bg-only, action swap) from the Kinetics400 dataset comprise the same amount of video data? Or are they just subsets? In this case, what's the amount of videos overall and per class of the new distilled datasets?

- For the action swap, how do you make up for camera motion (or align it) in the original video as well as in randomly chosen background videos?

- Why exclude larger views >50% of pixels? Wouldn't that exclude e.g. applying make-up or other actions which are usually recorded in a close-up.

Indicate pertaining:
It is hard to draw any conclusion without knowing which backbones rely on which pertaining. Please add this information to the text and (ideally) tables.

Human Action vs. Background for Human-Object Interaction
Another problem is revealed in Tab 4 which shows that using only the human might make limited sense in case of human object interactions and when it comes to tool usage. Here it would be good to open the discussion if things like screw drivers are part of 'using screwdriver' or not and how to capture that.
On a side note, I'm also not clear why scuba diving is not recognized. Shouldn't the diving suit give enough information here?

The relevant person might be to small
As all human figures are segmented, including bystanders that perform different actions, this might be an additional distractor. Would there be a good way of handling that?



Minor:
- Fig. 2 might profit from having the same scale for all four plots for easier comparability.

- It might be good to have some qualitative results in the main paper.

**Additional Feedback:**

Please see the questions under weakness and clarify how and which data you plan to release.

Justification of final rating: As there was no response by the authors and there is no clear idea how the data would be made available (which is obviously the main purpose of the track), I lower my score from 7 to 4.

Justification of final, final rating: After detailed discussion and the release of the toolkit and the respective validation videos, I agree with the other reviewers that this package has value for the community and I'm a happy to raise my score to the initial value.

**Clarity:**

The paper should be more explicit about the size of the newly generated data if it deviates from the original Kinetics-400 size.

**Correctness:**


The claims made in the paper look correct and valid. The proposed metrics are sound and make sense for the task at hand.

**Documentation:**


As the proposed new data is a derivative of a well know dataset, it should be fairly clear. For everything else, please see the questions under weakness.

**Ethics:**


Technically, Kinetics does, similar to Imagenet, not have consent from anyone recorded but might run under a guarantee of continued existence as it is a well-established dataset and therefore falls under the same category as many other vision datasets like ImageNet, MSCoco, and many more. As the proposed work is a derivative of Kinetics, any regulations that would apply to other datasets might also be assumed in this case.

**Relation To Prior Work:**

There are several references missing with regards to two aspects:

1) A similar idea has been tried for synthetic data, e.g. SURREACT: Synthetic Humans for Action Recognition from Unseen Viewpoints, Gül Varol, Ivan Laptev and Cordelia Schmid, Andrew Zisserman, IJCV 2021 and SURREAL, Learning from Synthetic Humans, Varol et al. CVPR 2017. Perhaps there might be more work in the synthetic or AR action area that I might have missed.

2) Discussing the role of background has a long tradition in action recognition actually. It would have been good to cover the discussion a bit, e.g.:
- Human Action Recognition without Human, Yun He et al., ECCVW 2016, https://openreview.net/forum?id=ryeujDzo
- Why Can't I Dance in the Mall? Learning to Mitigate Scene Bias in Action Recognition, J. Choi, NeurIPS 2019
- CATER: A diagnostic dataset for Compositional Actions and TEmporal Reasoning, Rohit Girdhar and Deva Ramanan, ICLR 2020
- and many more ...


**Summary And Contributions:**

The paper describes a new approach to address the problem of capturing the impact of visual background cues for action recognition.
To this end, the paper relies on recent advances in image/video segmentation and inpainting to segment the human figure in each frame. This allows separating the person that performs the executed action from the background.
The paper proposes to use the two cues, background and the human figure, to test in three variations: Human-only, background-only, and action-swap.
The evaluation is done extensively for various backbones and shows the impact of background vs. action under various aspects.

---

> ### Author Response · Authors · 2022-08-24
> **Official Comment**
>
> Dear Reviewer 2PNT,
>
> Thank you for your thoughtful comments and suggestions. Apologies for the delay – we were doing a detailed revision to incorporate all the feedback from the reviewers and wanted to make sure that is finalized before responding. We have substantially improved the introduction and abstract (Reviewer 5jrL), added related work (Reviewer 2PNT), re-organized the methods for better understanding (Reviewer ggQs, 5jrL), improved the presentation of the figures and tables (Reviewer 5jrL), and included a summary of the findings and the suggestions (Reviewer Usck, 5jrL).
>
> The toolkit is now available at https://github.com/princetonvisualai/HAT. The updated paper is uploaded.
>
> Here are the answers to some of the other questions that you have mentioned. We hope you will reconsider your final score (since we are still within the allotted rebuttal window).
>
> **Weaknesses:**
> - Re: The size of the augmented dataset
>
> We clarify in lines 170-172.
>
> For Human Only Videos and Background Only Videos, the size of the dataset is as same as the original dataset (Kinetics-400: 19877 videos, UCF101: 3783 videos)
>
> For Action Swap, we end up with smaller datasets due to the human coverage limitations.
> (Kinetics-400: 19,877 videos, UCF101: 3,783 videos) -> (Kinetics-400: 5,631 videos, UCF101: 1,694 videos)
> However, we use 3 random seeds, resulting in 16,893 (5631*3) action swap videos for Kinetics-400. It is possible to generate more action swap videos if desired.
>
>
> - Re: For the action swap, how do you make up for camera motion?
>
> We have added the details of the Action-Swap in the supplementary material (added in Section 9 of supplementary material). For Action-Swap, we align the first frames to have the same human segmentation weight center for both action frames and background frames. We use the same alignment we calculated in the first frame throughout the rest of the frames. This is done to keep the human body movement relative to the camera window. E.g., if a person moves from left to right in the camera window in the original video, we want to keep the same movement when the background is swapped.
>
>
> - Re: Why does Action Swap only take human coverage of 5~50%
>
> We want to “ensure that sufficient human and background cues are present in every generated Action-Swap video” (added in lines 167-168). The exact threshold was chosen through qualitative inspection. Please check (https://github.com/princetonvisualai/HAT/blob/main/doc/review_discussion.md#percentage-of-human-mask) to see the video frames with different human mask ratios.
>
>
> - Re: Indicate pre-training
>
> We have included the pretraining information in the main paper. (Table 1-2, Table 2 in the supplementary material)
>
>
> - Re: Human-object interaction is not well studied in our method.
>
> Human-Object Interaction is another exciting direction that we wanted to explore, but cannot be studied with the current toolkit, as our toolkit is specialized in the human body only. We leave the human-object interaction understanding to be future work. We now explicitly discuss this limitation in lines 123-128.
>
>
> - Re: Why is scuba diving not recognized?
>
> We believe that scuba diving is not well recognized because the model may use the surroundings (deep underwater with a dark blue tone) to make accurate predictions.
>
>
> - Re: Bystanders are included in the segmentation.
>
> Bystanders (people that are not participating with the labeled action) being included in the video is a known issue in video action recognition datasets. The issue of Kinetics-400 is more severe than UCF101, NTU-RGB+D, FineGym, and HAA500, where these datasets often have fewer bystanders.
> With Kinetics-400 not having a spatial annotation for specifying the person-of-interest, excluding the bystanders will be difficult. However, that was not the major problem in our case, as our toolkit is focused on human analysis, regardless of being a bystander or not.

---

> > ### Author Response · Authors · 2022-08-24
> > **Official Comment**
> >
> > **Other Comments:**
> >
> > - Re: Scale for all 4 plots in Figure 2
> >
> > We kept the scale to be different between the plots as the comparison is done within each plot, not between plots.
> >
> > - Re: Qualitative Results in the main paper.
> >
> > While we offer Figure 1 for qualitative examples, due to the page limit, we can not add any other qualitative results to the main paper. We have included model prediction on Action Swap videos in the supplementary material. (Figure 5 in the revised submission)
> >
> >
> > - Re: Explicit size of the newly generated data should be included.
> >
> > We have included the explicit size of the data (lines 171-173 in the revised submission). Thank you for the suggestions.
> >
> >
> > - Re: Missing references regarding the synthetic data and the role of the background.
> >
> > Thank you for the suggestion. We have modified the related works so that the paper includes deeper insights regarding both topics. In particular, we discuss synthetic data along with human action recognition datasets, including new references [53, 55, 56]. We discuss the background bias and out-of-context in the introduction and the additional related works in the supplementary material with new references [supplementary 15,6,17].
> >
> > [55] SURREACT: Synthetic Humans for Action Recognition from Unseen Viewpoints, Gül Varol, Ivan Laptev and Cordelia Schmid, Andrew Zisserman, IJCV 2021
> > [56] SURREAL, Learning from Synthetic Humans, Varol et al. CVPR 2017
> > [53] Render for CNN: Viewpoint estimation in images using CNNs trained with rendered 3d model views, Hao Su et al. ICCV 2015
> > [supplementary 15] Human Action Recognition without Human, Yun He et al., ECCVW 2016
> > [6] Why Can't I Dance in the Mall? Learning to Mitigate Scene Bias in Action Recognition, J. Choi, NeurIPS 2019
> > [17] CATER: A diagnostic dataset for Compositional Actions and TEmporal Reasoning, Rohit Girdhar and Deva Ramanan, ICLR 2020

---

> > > ### Comment · Reviewer_2PNT · 2022-08-26
> > > **Response to authors comment**
> > >
> > > Dear authors,
> > >
> > > thanks so much for the clarification. I have to say that the response raises more questions than it answers.
> > >
> > > My biggest concern is the deviation in size. The original Kinetics 400 dataset has been released with 306K video clips ( https://arxiv.org/abs/1705.06950 ) . Even given that 10%-15% might be gone over the years, your Kinetics version roughly covers 20K, and the action swap results are executed on a scale on HMDB + UCF . I don't think that those numbers justify the claim that action swap or influence of background is tested at a large scale.
> > >
> > > Also, how do you train for 400 action classes on 20K samples? That would mean 50 training samples per class (less than UCF or HMDB on 4x more classes). I have a hard time understanding how you can e.g. reach 73.57% accuracy on I3D with imagenet initialization under those conditions (or any other model in Tab 1). Just as an example, the original I3D paper reports 71.1 (RGB) 63.4 (OF) 74.2 (RGB+OF) on the original Imagenet + Kinetics dataset (400 samples per class min., https://arxiv.org/pdf/1705.07750.pdf). I understand that training has become more sophisticated since then, but I have a hard time getting that they can make up for such a drop in training data. Do you use the standard validation set?
> > >
> > > About the framework, I agree that it would be good to have, but without the data, I don't think anyone would reasonably apply it for evaluation + I would even question the energy/value tradeoff of such an exercise.
> > > I understand that Kinetics is not easy in terms of licensing, but given the numbers, you would not have lost too much by running this on a combination of free datasets (UCF, HMDB, Charades, various sports datasets like Multi-sport, Diving 48, etc ... ). So as the Kinetics licensing issue is known, why not plan directly for something that could be open-sourced and that could have real value for the community?
> > >
> > > As a conclusion, I don't think that this project in its current state raises to many questions, also with respect to its value to the community, and should not be published.

---

> > > > ### Author Response · Authors · 2022-08-26
> > > > **Comment**
> > > >
> > > > Dear Reviewer 2PNT,
> > > >
> > > > We want to clarify that our work is an **evaluation benchmark**, not a training dataset. All our experiments are done on released trained models, which are trained on the unmodified original Kinetics-400 training dataset. We only use our generated videos to evaluate models to see which visual cues they are relying on when making predictions. The size of our synthetic evaluation dataset is as same as the Kinetics-400 validation set. We understand the confusion on the original submission as Reviewer 5jrL as noted as well, and the current revision is more clear that our work is a framework for manipulating the validation data (Section 3.2) and providing metrics (Section 3.3) for quantitatively analyzing background bias and human understanding of **trained human action recognition model**. Please let us know if our revised introduction and abstract is more clear.
> > > >
> > > > - Re: Training and accuracy
> > > >
> > > > We want to reemphasize that we do not train any models (lines 204-205), so the original accuracy (accuracy on unmodified Kinetics-400) is similar to the reports in the paper or the number reported by MMAction2. E.g., on I3D we see 73.57% (Our Paper), 71.1% (I3D paper), and 73.27% (MMAction2). In Table 1, column “BAcc” tabulates our proposed metric, i.e., accuracy when using our **synthetic validation set** using the same trained model. Throughout the paper, we discuss the usage of the proposed metric and meaningful findings from evaluating 74 trained models.
> > > >
> > > > - Re: Application
> > > >
> > > > We address the need for our work in the introduction (lines 16-29) and related work (lines 83-101). We explain that despite many works of human action recognition that work on background debiasing, they do not use a clear quantitative metric on background bias. We believe that we offer a quantitative analysis of background bias that can benefit the community.
> > > >
> > > > - Re: Energy/value trade-off
> > > >
> > > > While we admit that generating the evaluation dataset requires initial cost (lines 208-210), given that this is an evaluation dataset, we see that our method can be more energy efficient (lines 308-311) compared to existing methods of human-centric analysis which requires, e.g., fine-tuning a model to a specific dataset like Diving48.
> > > >
> > > > - Re: Release of our synthetic dataset and other datasets.
> > > >
> > > > We have uploaded the generated synthetic videos and have released them. Please check the supplementary material (line 92). Our generated dataset is based on the publicly released version of the Kinetics-400 validation set from CVDF (https://github.com/cvdfoundation/kinetics-dataset).
> > > >
> > > > We have included the experiment using UCF101 trained models in the supplementary material. We plan to upload the generated dataset and attach the link in our GitHub repository. Please note that most of the datasets that the reviewer mentioned have similar license issues with Kinetics. UCF101 (YouTube), HMDB51 (movies, youtube, etc.), Diving48 (web download), and multi-sport (YouTube).

---

### Official Review · Reviewer_ggQs · 2022-07-26
**Review of "Enabling Detailed Action Recognition Evaluation Through Video Dataset Augmentation"**

**Rating:** 6
**Confidence:** 4

**Strengths:**

- Novel method to quantify the influence of background using Human-Only Ratio (HOR), Background-Only Ratio (BOR), and SHAcc and SBErr on action swap methods (same, random, close, far)
- Combined image segmentation and video inpainting to generate background-only, human-only and action-swap dataset
- Provided new metrics on top of accuracy which would be useful to quantify the influence of background on model predictions.
- Provided some valuable insights into design choices that can improve or worsen the influence of background on model predictions based on their evaluation using BOR and HOR i.e. activity recognition models lean towards using background to predict action and high accuracy models tend to focus on human better than other models.


**Weaknesses:**

- While one of the main highlights of the paper was that it introduces a human-centric Analysis Toolkit, it is confusing as there is no public link nor code provided in the supplementary materials at this point of review.
- The paper analyzes 74 trained activity recognition models, but there were only a total of 19 architectures trained on different dataset, backbones, pretraining, resolution, sampling, clip length, stride, number of clips. Only the effect of dataset, sampling, and clip length were discussed. The number of clips, backbone, pretraining and resolution might also play a part in model performance and should be looked into.
- Table 1 and 2 compared models from different backbones i.e. ResNet50, ResNet101, ResNet152, TimeSformer which might not provide a far comparison. Backbone seems to play a part as models with better performance were also trained on larger backbones.
Table 3 compared the different models but details about their backbone/ pretraining were not provided.
- I cannot find any explanation of normal-scale dataset and large-scale dataset even though they were analysed in section 4.4 and used in Table 1, Table 2 and Figure 3 in the main paper and Table 3-6 in the supp. materials.

**Additional Feedback:**

- It is a strong submission and provides a way to quantify the longstanding problem of the influence of background in human-centric problems.
- Additionally, the paper could provide suggestions on how to combat the problem of background reliance from a data perspective i.e. train on their augmented action-swap dataset or collect better data with more diverse scenes?
- The paper has some useful and interesting insights. Perhaps they could be summarised to give some recommendations on how to train a more robust activity recognition model i.e. use more frames for training, adopt TSM design that could exploit temporal difference to improve HOR etc
- The modified Kinetics-400 was provided in the supplementary materials. I think it could serve as a new benchmark dataset to evaluate the different metrics. This can be added into the main paper as a feature.


**Clarity:**

- In general, it is a very well written and concise paper
- I think Swap Human Accuracy (SHAcc) and SBErr (SBE) should be added and explained in section 3 under methodology. The first mention in Table 3 was slightly confusing.
- While the paper analyzes 74 trained activity recognition models, it is slightly confusing as only a few architectures were discussed in the main section. Table 3 in the supplementary materials showed that there were only 19 architectures but the model variants were trained on different dataset (normal-scale and large-scale), backbones, resolution, sampling, clip length, stride, number of clips, totaling 74 models. Perhaps some of the details about the models in the supplementary materials could be moved to the main section for clarity.


**Correctness:**

- I noticed that action swap will lead to unrealistic scenes i.e. floating face/ person in demo videos and I am concerned if that might affect accuracy.
- I looked through the provided dataset and in some background-only videos, the person is still visible. This might affect the evaluation for BOR. Perhaps the paper could also mention this as a limitation i.e. the robustness of the dataset also depends on the accuracy of the image segmentation and in-painting model.
- L257-261 “using a larger dataset just to improve accuracy is not an ideal training strategy” seems like a large generalization. What characteristics of the large scale dataset makes it not ideal? I.e. are there too many duplicates of similar backgrounds?
- L260-261  “background fixation of the model became worse when using a large-scale dataset incorrectly” - This seems to be a rather bold statement given (1) there is no explanation of what constitutes a large-scale dataset in the paper (2) there is only a single example given. Perhaps more experiments could be run to support this claim. The authors could also expound on how the large scale dataset is used incorrectly? And conversely, how to use it correctly?


**Documentation:**

- Some concerns of how the toolkit will be distributed and possibly maintained since no github/ public link was provided.
- Currently only a jupyter notebook is provided to show some visualisation. A link to the toolkit and instructions on how to use it could be added into the submission. In addition, it would be nice to have an evaluation pipeline for the metrics.


**Relation To Prior Work:**

- For sections 4.2-4.4, there are quite a number of insights. More literature could be reviewed to see if other works might agree/ concur with the current observations, or if the observations are new and not discussed previously.

**Summary And Contributions:**

This paper:
- proposed a novel method to quantify the influence of background on model predictions
- proposed several metrics i.e. Background-Only Ratio (BOR), Human-Only Ratio (HOR), Swap Human Accuracy (SHAcc), Swap Background Error (SBError) in addition to accuracy
- benchmarked a variety of models (74 trained activity recognition models) on an augmented dataset and provide insights about what makes certain models robust to background
- studied the impact of scene background on action recognition

---

> ### Author Response · Authors · 2022-08-24
> **Official Comment**
>
> Dear Reviewer ggQs,
>
> We thank you for your detailed review. We answer some of your questions below.
>
>
> **Weakness:**
> - Re: Toolkit is not released
>
> We have now released the toolkit (https://github.com/princetonvisualai/HAT).
>
>
> - Re: the effect of all the different hyperparameters (backbones, pretraining, etc)
>
> We have modified Tables 1-2, and supplementary material Table 2, and added the backbone/pretraining information so that the comparisons can be made more directly. Thank you for the suggestion. We limit our analysis to the 74 different trained models using weights offered by MMAction2 (we clarify this fact in line 44 in the introduction). Thus, we are not able to always control for all the different hyperparameters but do the best we can (added discussion in lines 205-207). However, the HAT framework can certainly be used by researchers in the future on their new models, to more directly analyze the design choices.
>
> - Re: What are normal-sized and large-sized settings?
>
> Normal-scale dataset labels the trained weights that had only access to strongly supervised datasets like ImageNet, Kinetics400. Ones labeled Large-scale dataset had access to IG-1B, IG65M, Sports1M, and OmniSource. We have included this in the revised version of the paper. (added in lines 213-214)
>
> **Correctness:**
> - Re: Unrealistic (Synthetic) video impacting the performance.
>
> Using synthetic data to understand the model behavior is widely used in the image domain. Notably, [1,2,3] used a pasted object in an image to study the object recognition/detection model, and [4,5,6] use a grid-like image on image recognition models. Following these works, we propose using synthetic video to evaluate trained models.
>
> Nevertheless, your point is well-taken. One result which might partially mitigate this concern is noting that the human action recognition accuracy on Same Action-Swap videos (where the human figure segmented from one video is pasted onto the background from a different video but from the same action class) is not dramatically dissimilar from the original human action recognition accuracy (evaluated on real videos). For example, the SlowFast ResNet101 model has Same Action-Swap accuracy of 69.6% (Table 2, “Same” column) compared to the original accuracy of 78.1% (Table 1, “OAcc” column).
>
> [1] The elephant in the room, Amir Rosenfeld, et. al., arXiv, 2018
> [2]  Strike (With) a Pose: Neural Networks Are Easily Fooled by Strange Poses of Familiar Objects, Michael A. Alcorn, et. al., CVPR, 2019
> [3] Noise or Signal: The Role of Image Backgrounds in Object Recognition, Kai Yuanqing Xiao, et. al., ICLR 2021
> [4] Convolutional Dynamic Alignment Networks for Interpretable Classifications, Moritz B ̈ohle, et. al., CVPR 2021
> [5] Towards Better Understanding Attribution Methods, Sukrut Rao, et. al., CVPR 2022
> [6] B-cos Networks: Alignment is All We Need for Interpretability, Moritz B ̈ohle, et. al., CVPR 2022
>
> - Re: Person is still visible in BOR. This can be added as a limitation.
>
> Thanks for pointing out the issue. We have included it in the limitation section (lines 284-285 in the original submission, lines 306-307 in the revised submission).
>
> - Re: Claims in the large-scale dataset.
>
> Thank you for the detailed suggestions and questions, and we have re-written the section. We have improved the section (lines 281-286 in the revised submission), and it now includes a fair comparison between different trained weights where all the settings are the same except for the pre-training. The previous revision (lines 254-261 in the original submission) included comparisons between different settings, which are now fixed. We are sorry for the confusion that it made. The new study shows that the large-scale dataset pre-training helps the original accuracy and Swap Human Accuracy, but increases in background error as well. This shows that the large-scale dataset pre-training improves better feature detecting, but seems to have less or no impact on better human focusing.
>
> **Clarity:**
>
> - Re: SHAcc and SBErr should be added to section 3.
>
> We introduced the SHAcc and SBErr in Section 3 (added in lines 182-188). Thank you for the suggestion
>
>
> - Re: few Architectures were discussed in the main section.
>
> We have revised the paper so that the choice of the presented weights is more consistent. Among trained weights that share the same model architectures and the backbone, we select the setting where the original accuracy is the highest (added in caption of Table 1). Such restriction is only on the tables. Figures 2-3 do not have restrictions and Figure 3 shows all 74 models.
>
> We have included the backbone and the pre-training in Table 2 (Table 3 in the original submission). Thank you for the suggestion.

---

> > ### Author Response · Authors · 2022-08-24
> > **Official Comment**
> >
> > **Other feedbacks:**
> > - Re: Release plan and evaluation pipeline.
> >
> > We have released the codes for the toolkit we used (https://github.com/princetonvisualai/HAT). The toolkit also includes the evaluation pipeline that can be used in MMAction2.
> >
> >
> >
> > - Re: Lack of recommendation for future model design.
> >
> > Thank you for the suggestion. We have provided the recommendation for future model design in the conclusion (added in lines 298-304)
> > “From our findings, we suggest the future researchers to (1) not rely on the accuracy as the only metric, as original accuracy do not fully represent the performance of the model based on the human action; (2) carefully select the temporal hyper-parameters, as temporal parameters can improve/worsen the background bias of human action recognition models; and (3) use HAT toolkit to see if the model design (e.g., as Non-local) can improve your model on accuracy and reduce the background bias. We hope that this tool can be adopted by future researchers for a better human-centric analysis of human action recognition models.”

---

> > > ### Comment · Reviewer_ggQs · 2022-08-26
> > > **Post-rebuttal**
> > >
> > > Thanks for the responses, my concerns are basically addressed. I appreciate the clarification and revisions from the authors.

---

### Official Review · Reviewer_Ep25 · 2022-07-26
**Review of Enabling Detailed Action Recognition Evaluation Through Video Dataset Augmentation**

**Rating:** 7
**Confidence:** 4

**Strengths:**

The manuscript is very easy to read, and this reviewer clearly enjoyed reading it. Also, the findings (e.g. Lines 226-233) are interesting.

The authors demonstrated the validity of their proposed models on various video model architectures (followed by the MMAction2 [6] implementation).


**Weaknesses:**

One of the weaknesses of this paper is that it is unclear what the contributions of this paper are to the datasets or benchmarks. The authors invented the toolkit using the two existing models for image segmentation and video inpainting. However, combining these two approaches is not enough for a solid contribution. Though it is interesting to see the human-centric analysis of existing models using the toolkit, the authors' plan for this benchmark (or datasets) is not specifically mentioned in the manuscript. In Lines 168-169, the authors said the swap pairs would be released. Would the authors also release the toolkit?

Some other limitations are also written in the sections below.

**Additional Feedback:**

In Tables, I wonder if * can be replaced with some other character that might be middle-aligned instead of top-aligned. A better suggestion might be to show the statistics of the different settings by showing std.

In Figure 3 (left), some dots are not covered with black lines.

**Clarity:**

Figure 1 Right and Section 3.4: Are Random/Close/Far/Same related to the data distribution? If yes, how do the authors make the distinctions between the three categories? For example, in Line 149, the authors defined "Close" when the background is swapped with a video from a class that has a similar background. In line 160, it is more specified with "low L1 distance" but is still vague with "low".

In Lines 157-164: the authors computed the average prediction probability of each class. Does the class indicate an action category or a scene category? If that refers to an action category, averaging scores per class relies on the assumption that videos in each class will share a similar scene, which could be correct but not always. This reviewer suggests computing the background similarity based on individual scene scores instead of average scene scores per action category.

In Line 156, why do the authors only keep the videos with all the frames having human masks taking 5-50% of the pixels? That makes the benchmark unnatural and less interesting. What percentage of the videos satisfy the criterion? Did the authors use the subsets for testing the models in the results section?

**Correctness:**

In Lines 179-180, the authors said they did not train any models but tested the existing models on the newly generated videos. While the frames shown in Figure 1 look natural, this reviewer is curious whether the original models show low performances on their datasets (Background Only Videos, Human Only Videos, Action Swap Videos) due to some artifacts. Namely, the models haven't been exposed to the synthesized video frames. As an example, HAcc performances are relatively lower than BAcc performances. This reviewer wonders if the percentage of the synthesized pixels also matters to the performance.

**Documentation:**

One of the weaknesses of this paper is it is not clear about the data (or benchmark) release plan. Further, there are no details about the HAT.

**Ethics:**

The authors talked about the ethical concerns in the discussion session.

**Relation To Prior Work:**

The authors showed the related work in section 2 by clarifying the distinction between the proposed work.

**Summary And Contributions:**

In this paper, the authors introduced a new toolkit for human action evaluation on an existing dataset. In particular, their tool generates synthetic video frames by using image segmentation models to segment the boundary of human regions and using a video-inpainting method to fill the gap of the pixels in human boundaries. Then they evaluated various existing models to validate how much existing models are influenced by background rather than human-related regions. The authors also claimed that the models trained on longer time sequences are less dependent on the scene but rather focused on the human actions.

---

> ### Author Response · Authors · 2022-08-24
> **Comment to Reviewer Ep25**
>
> Dear Reviewer Ep25,
>
> We appreciate your reviews and suggestions that can help us improve the paper. Here we address the issues you have mentioned.
>
> **Weakness:**
> - Re: Unclear if the contribution is the dataset or benchmark
>
> We updated the first paragraph of Section 3 to read: “Our Human-centric Analysis Toolkit (HAT) is a general framework that can be used to measure the amount of background bias learned by a human action recognition model. HAT takes two inputs: (1) a trained human action recognition model and (2) a set of validation videos each annotated with the human action class. HAT then proceeds in three steps. First, it leverages human segmentation models to separate the human visual cues from the background visual cues in the validation videos. Second, it generates six sets of counterfactual validation videos, including Human-Only, Background-Only, and four sets of Action-Swap videos (see Figure 1 for examples). Finally, it evaluates the trained model on these counterfactual videos and returns a set of ten metrics which quantify the different effects of background bias. This methodology can expand the dataset without any need to manually collect new data, allowing deeper analysis of human action recognition in an affordable manner.”
>
> We would welcome further feedback.
>
> - Re: Combining Inpainting and Segmentation is not enough for a solid contribution.
>
> We revised Section 3 to better highlight the contributions and challenges of designing our toolkit. While the toolkit relies on existing image segmentation and video inpainting models, the intellectual contribution is in the different types of generated counterfactual videos (Section 3.2, particularly the Action-Swap videos of Section 3.2.1), as well as the metrics (Section 3.3). Given that this is the first work that can quantitatively measure the effect of human cues on human action recognition models, along with exhaustive experiments evaluating 74 different models across 25 different model architectures, we believe this work to be a solid contribution to the video action recognition community.
>
> - Re:  Plan to release the toolkit
>
> We have released the toolkit (https://github.com/princetonvisualai/HAT).
>
> **Correctness:**
> - Re: Unrealistic (Synthetic) video impacting the performance.
>
> Using synthetic data to understand the model behavior is widely used in the image domain. Notably, [1,2,3] used a pasted object in an image to study the object recognition/detection model, and [4,5,6] use a grid-like image on image recognition models. Following these works, we propose using synthetic video to evaluate trained models.
>
> Nevertheless, your point is well-taken. One result which might partially mitigate this concern is noting that the human action recognition accuracy on Same Action-Swap videos (where the human figure segmented from one video is pasted onto the background from a different video but from the same action class) is not dramatically dissimilar from the original human action recognition accuracy (evaluated on real videos). For example, the SlowFast ResNet101 model has Same Action-Swap accuracy of 69.6% (Table 2, “Same” column) compared to the original accuracy of 78.1% (Table 1, “OAcc” column).
>
> [1] The elephant in the room, Amir Rosenfeld, et. al., arXiv, 2018
> [2]  Strike (With) a Pose: Neural Networks Are Easily Fooled by Strange Poses of Familiar Objects, Michael A. Alcorn, et. al., CVPR, 2019
> [3] Noise or Signal: The Role of Image Backgrounds in Object Recognition, Kai Yuanqing Xiao, et. al., ICLR 2021
> [4] Convolutional Dynamic Alignment Networks for Interpretable Classifications, Moritz B ̈ohle, et. al., CVPR 2021
> [5] Towards Better Understanding Attribution Methods, Sukrut Rao, et. al., CVPR 2022
> [6] B-cos Networks: Alignment is All We Need for Interpretability, Moritz B ̈ohle, et. al., CVPR 2022
>
>
> - Re: Percentage of the synthesized pixels to the performance.
>
> We study the percentage of the synthesized pixels on accuracy: (https://github.com/princetonvisualai/HAT/blob/main/doc/review_discussion.md#percentage-of-synthetic-pixels). As expected, in all 3 experiments, videos that have more synthetic pixels do tend to have lower accuracy. However, such tendency is the least in Action Swap experiments, which we designed specifically to mitigate this concern.

---

> > ### Author Response · Authors · 2022-08-24
> > **Comment to Reviewer Ep25 - 2**
> >
> > **Clarity:**
> > - Re: Selection of ActionSwap videos
> >
> > We revised the manuscript to clarify; please see the new Section 3.2.1. The particularly relevant paragraph is:
> >
> > “To determine the appropriate classes for Close and Far Action-Swap videos, we need to determine how similar the backgrounds are across different classes. To do so, we first feed the frames from the original validation videos into a Places365 [72] trained scene classification model. For each action class, we then compute the average scene prediction vector by averaging the prediction probabilities from all frames of all videos of this class. We can then rank all the other classes according to the L1 distance in their average scene prediction vector. We consider the class to be “close” if it’s among the 5 classes with the smallest L1 distance and “far” if it’s among the 200 largest (of 399 classes total).”
> >
> > For generating the Far Action-Swap videos, we need to find an action class with overall dissimilar scenes rather than identifying individually-dissimilar scenes for every video. Consider, for example, the action class “eating,” which can occur in many different scenes: e.g., kitchens, restaurants, picnic tables outside, by a campfire, etc. Swapping the background from a single seemingly-dissimilar individual video might result in a synthetic example with actually a plausible background. In contrast, our approach would swap in backgrounds from action classes like “gymnastics” (which occur in gyms, where people in fact rarely eat).
> >
> > - Re: Why does Action Swap only take human coverage of 5~50%
> >
> > We want to “ensure that sufficient human and background cues are present in every generated Action-Swap video” (added in lines 167-168). The exact threshold was chosen through qualitative inspection. Please check (https://github.com/princetonvisualai/HAT/blob/main/doc/review_discussion.md#percentage-of-human-mask) to see the video frames with different human mask ratios.
> >
> >
> > - Re: Size of the Action Swap dataset
> >
> > For Action Swap, we end up with smaller datasets due to the human coverage limitations. (added in lines 171-173)
> > (Kinetics-400: 19,877 videos, UCF101: 3,783 videos) -> (Kinetics-400: 5,631 videos, UCF101: 1,694 videos)
> > However, we use 3 random seeds, resulting in 16,893 (5631*3) action swap videos for Kinetics-400. It is possible to generate more action swap videos if desired.
> >
> > **Additional Feedback:**
> >
> > - Re: Table presentation regarding averaged values.
> >
> > We have decided not to average the values in the table, as it could be misleading to average between different settings. Instead, we have shown the most representative setting (one with the highest original accuracy) per each backbone and the pre-training (added in the captions of Table 1).
> >
> > - Re: In Figure 3, some dots are not covered with black lines:
> >
> > As written in the caption we have bordered the dots that belong to TSM and TSN (in the captions of Figure 3). This is for better visibility as TSM and TSN are compared in section 4.3.

---

> > > ### Comment · Reviewer_Ep25 · 2022-08-29
> > > **Post-rebuttal**
> > >
> > > Thanks for the responses and revising the paper. Most of my concerns have been resolved with the responses. So, I updated my rating to 7.

---

### Official Review · Reviewer_Usck · 2022-07-27
**Benchmarking video action recognition models**

**Rating:** 7
**Confidence:** 4
**Correctness:** I don't have a reason to doubt the co…
**Clarity:** Yes

**Strengths:**

*The presented work is an interesting area and the article is reasonably well written
*The benchmarking results are thorough and supplementary information provides good context that can be used to choose different models
*Important observation that existing models learn based on background

**Weaknesses:**

*While the benchmarking is thorough, it uses only Kinetics-400. Some discussion about other datasets such as Moments in Time, VLOG would have been a good addition.
*Not much discussion that could be used to help researchers chart out new problems to solve using their results
*Many of the tools described in this article have already been released and/or published

**Additional Feedback:**

None

**Documentation:**

Most of the work is self contained and documentation of related tools is clear.

**Relation To Prior Work:**

Yes

**Summary And Contributions:**

The authors present their work on analyzing various action recognition models. Specifically, they present a toolkit that can be used to evaluate action recognition models, an evaluation of 74 released video action recognition models and design choices that can be used to improve action recognition models.

While the article was interesting, my large concern was in the lack of discussion/summarization of the results which could be used to further research in this area.

---

> ### Author Response · Authors · 2022-08-24
> **Comment to Reviewer Usck**
>
> Dear Reviewer Usck,
>
> Thank you for your thoughtful comments and for being supportive of our work. We comment on your concerns below.
>
> - Re: limited discussion/summarization of results
>
> Yes, absolutely, we made a number of revisions to address this concern. Please see lines 45-56 in the new introduction where we highlight the key findings. To summarize briefly: (1) “All 74 models exhibit strong background bias”, (2) “Models trained with fewer frames per video appear to be more prone to background bias”, and (3) “Models trained with dense temporal sampling around a single timestep appear to be more prone to background bias compared to models trained with uniform sampling throughout the video.” We include more details in the paper.
>
> We also incorporated more discussion of our findings throughout. For example, we included a discussion of the temporal/spatial modeling in lines 275-280, and of the use of large-scale datasets for pre-training in lines 281-286. Finally, we summarize the findings in lines 291-297, and we conclude with suggestions for future researchers in lines  298-304.
>
> - Re: only using the Kinetics-400 dataset.
>
> We use Kinetics-400 as our primary test bed because almost all human action recognition models have the weights released for Kinetics-400 (line 172 in the original submission, lines 190-191 in the revised version). However, to demonstrate the generality of our tool, we additionally included results on UCF101 in the supplementary material (Section 4 in the original submission, Section 5 in the updated version). Our toolkit can be used on any human action dataset (mentions in lines 194-195), including VLOG, but is not suitable for Moments in Time, as the MiT dataset is not focused on human action.
>
>
> - Re: Using released/published tools
>
>
> Right, our toolkit leverages existing segmentation and inpainting tools. Our novelty lies in using these existing methods to design the first toolkit that quantitatively evaluates the effect of background on human action recognition models. We revised Section 3 to better highlight the contributions and challenges of designing our toolkit. While the toolkit relies on existing image segmentation and video inpainting models, the intellectual contribution is in the different types of generated counterfactual videos (Section 3.2, particularly the Action-Swap videos of Section 3.2.1), as well as the metrics (Section 3.3).
>
> We also expanded the supplementary material (Section 1.1, 1.2) to further detail the importance of using modern impainting and segmentation models as opposed to simpler baselines. The supplementary provides the need of using video inpainting, where the naive baseline does not provide the same results as the inpainted experiments (lines 19-26 in supplementary materials).  We also show the importance of using temporally robust segmentation by showing missing segmentations can lead to worse inpainting. Without these modern methods, our toolkit would not have been possible.

---

### Review · Ethics_Reviewer_iWMp · 2022-08-20

**Recommendation:** 1

**Ethics Documentation:**

The authors describe how the data was collected—from other datasets that have licenses that permit derivative works. The authors' handling of data is consistent with norms in the international ML community.

The authors do not state how the new dataset will be licensed.

The authors do not state how the data will be made available—they only promise it will be and provide a temporary Google drive link.

**Ethics Review:**

This paper was flagged for ethics review due to a concern about licensing—specifically the fact that dataset created is a derivative of existing datasets that may not have consent for images of people that they contain.

The authors are US-based and consent for likenesses is not required in the US. The authors explicitly describe the license of the datasets they derive from, and these licenses permit the creation of derivative works.

The authors do not (as far as I can see) explicitly state the license for their new dataset. They should do this.

---

> ### Author Response · Authors · 2022-08-24
> **Ethics Review Comment**
>
> Thank you! Our code is now available at https://github.com/princetonvisualai/HAT. As our code is derived from other existing codes, please refer to the individual licenses (Apache 2.0 and CC BY-NC 4.0) when using our toolkit.
>
> The synthetic generated datasets are derived from the Kinetics-400 and can be re-generated using our code (we include the random seeds used in the paper). Our dataset admittedly suffers from the same licensing issues as Kinetics-400; some of the original YouTube videos included in Kinetics-400 do not specify any license. Our released dataset is a derived work of Kinetics-400 and follows the Creative Commons Attribution 4.0 International License for the annotation. For the synthetic video, please refer to the license of the individual videos which can be found at https://github.com/cvdfoundation/kinetics-dataset.

---

### Author Response · Authors · 2022-08-24
**Overall Comment**

Dear reviewers,

Thank you very much for the fruitful comments and detailed suggestions! We revised the paper and uploaded it above. Moreover, we have released the toolkit implementation (https://github.com/princetonvisualai/HAT) that we used to generate the synthetic videos and experimental results; we also included the random seeds to ensure full reproducibility. Please let us know if you have any further thoughts!

Thank you.

---

### Meta-Review · Area_Chair_wLs9 · 2022-09-10

**Recommendation:** Accept
**Confidence:** 4

**Metareview:**

The paper provides an evaluation toolkit and leaderboard for comparing current action recognition models on the famous Kinetics-400 dataset for their background bias. This is achieved by using publicly available training weights and carefully curating a validation set to quantitatively assess the impact of such bias.

Before the rebuttal, the reviewers were on the negative side due to writing confusion as well as the fact that the toolkit was not submitted with the first version.

Luckily to the authors, the reviewers were happy to engage actively with the revised version that seems to have resolved their concerns. They all recommend acceptance.

As an AC, I think the authors should have provided their toolkit with the first submission for thorough and detailed assessment, as this is their main contribution. But, in light of all reviewers recommending acceptance, I believe this is sufficient proof of the value to the research community and would thus recommend acceptance of this work as a poster.

---

### Decision · Program_Chairs · 2022-09-16

Accept